# EvenNet: Ignoring Odd-Hop Neighbors Improves Robustness of Graph Neural Networks

**Runlin Lei**
Renmin University of China
runlin_lei@ruc.edu.cn

**Zhen Wang**
Alibaba Group
jones.wz@alibaba-inc.com

**Yaliang Li**
Alibaba Group
yaliang.li@alibaba-inc.com

**Bolin Ding**
Alibaba Group
bolin.ding@alibaba-inc.com

**Zhewei Wei** [*]
Renmin University of China
zhewei@ruc.edu.cn

## Abstract

Graph Neural Networks (GNNs) have received extensive research attention for their promising performance in graph machine learning. Despite their extraordinary predictive accuracy, existing approaches, such as GCN and GPRGNN, are not robust in the face of homophily changes on test graphs, rendering these models vulnerable to graph structural attacks and with limited capacity in generalizing to graphs of varied homophily levels. Although many methods have been proposed to improve the robustness of GNN models, the majority of these techniques are restricted to the spatial domain and employ complicated defense mechanisms, such as learning new graph structures or calculating edge attention. In this paper, we study the problem of designing simple and robust GNN models in the spectral domain. We propose EvenNet, a spectral GNN corresponding to an even-polynomial graph filter. Based on our theoretical analysis in both spatial and spectral domains, we demonstrate that EvenNet outperforms full-order models in generalizing across homophilic and heterophilic graphs, implying that ignoring odd-hop neighbors improves the robustness of GNNs. We conduct experiments on both synthetic and real-world datasets to demonstrate the effectiveness of EvenNet. Notably, EvenNet outperforms existing defense models against structural attacks without introducing additional computational costs and maintains competitiveness in traditional node classification tasks on homophilic and heterophilic graphs. Our code is available in `https://github.com/Leirunlin/EvenNet`.

## 1 Introduction

Graph Neural Networks (GNNs) have gained widespread interest for their excellent performance in graph representation learning tasks [10, 13, 17, 26, 28]. GCN is known to be equivalent to a low-pass filter [2, 21], which leverages the homophily assumption that "connected nodes are more likely to have the same label" as the inductive bias. Such assumptions fail in heterophilic settings [37], where connected nodes tend to have different labels, encouraging research into heterophilic GNNs [1, 22, 37]. Among them, spectral GNNs with learnable polynomial filters [3, 9, 14] adaptively learn suitable graph filters from training graphs and achieve promising performance on both homophilic and heterophilic graphs. If the training graph is heterophilic, a high-pass or composite-shaped graph filter is empirically obtained.

---

[*]Zhewei Wei is the corresponding author. The work was partially done at Gaoling School of Artificial Intelligence, Peng Cheng Laboratory, Beijing Key Laboratory of Big Data Management and Analysis Methods and MOE Key Lab of Data Engineering and Knowledge Engineering.

36th Conference on Neural Information Processing Systems (NeurIPS 2022).

While GNNs are powerful in graph representation learning, recent studies suggest that they are vulnerable to adversarial attacks, where graph structures are perturbed by inserting and removing edges on victim graphs to lower the predictive accuracy of GNNs [38, 31]. Zhu et al. [36] first established the relationship between graph homophily and structural attacks. They claimed that existing attack mechanisms tend to introduce heterophily to homophilic graphs, which significantly degrades the performance of GNNs with low-pass filters. On the one hand, several attempts are made to improve the robustness of GNNs against the injected heterophily from the spatial domain [11, 16, 29, 34, 35]. These methods either compute edge attention or learn new graph structures with node features, requiring high computational costs in the spatial domain. On the other hand, while spectral GNNs hold superiority on heterophilic graphs, their performance under structural perturbation is unsatisfactory as well, which arouses our interest in exploring the robustness of current spectral methods.

In this study, we consider homophily-heterophily inductive learning tasks, which naturally model non-targeted structural attacks. We observe that structure attacks enlarge the homophily gap between training and test graphs besides introducing heterophily, challenging spectral GNNs to generalize across different homophily levels. Consequently, despite their outstanding performance on heterophilic graphs, spectral GNNs such as GPRGNN have poor generalization ability when the training and test graphs have different homophily. For example, suppose we now have two friend-enemy networks like the ones in Figure 1. If friends are more likely to become neighbors, representing the relationship "like", the network is homophilic. If enemies form more links corresponding to the relationship "hate", the network becomes heterophilic. If we apply spectral GNNs trained on "like" networks (where a low-pass filter is obtained) to "hate" networks, we will mistake enemies for friends on "hate" networks. Despite the strength of spectral GNNs in approximating optimal graph filters of arbitrary shapes, the lack of constraints on learned filters makes it difficult for them to generalize.

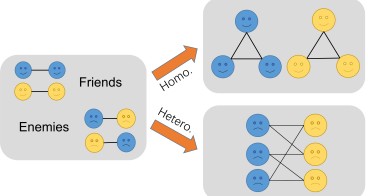

Figure 1: Two friend-enemy networks of opposite homophily.

To improve the performance of current spectral methods against non-targeted structural adversarial attacks, we design a novel spectral GNN that realizes generalization across homophily. Our contributions are:

- We proposed EvenNet, a simple yet effective spectral GNN that can be generalized to graphs of different homophily. EvenNet discards messages from odd-order neighbors inspired by balance theory, deriving a graph filter with only even-order terms. We provide a detailed theoretical analysis in the spatial domain to illustrate the advantages of EvenNet in generalizing to graphs of different homophily.

- We propose Spectral Regression Loss (SRL) to evaluate the performance of graph filters on specific graphs in the spectral domain. We theoretically analyze the relationship between graph filters and graph homophily, confirming that EvenNet with symmetric constraints is more robust in homophily-heterophily inductive learning tasks.

- We conduct comprehensive experiments on both synthetic and real-world datasets. The empirical results validate the superiority of EvenNet in generalizing to test graphs of different homophily without introducing additional computational complexity while remaining competitive in traditional node classification tasks.

## 2 Preliminaries

**Notations.** Let $\mathcal{G} = (\mathcal{V}, \mathcal{E})$ denote an undirected graph, where $N = |\mathcal{V}|$ is the number of nodes. Let $A \in \{0, 1\}^{N \times N}$ denote the adjacency matrix. Concretely, $A_{ij} = 1$ indicates an edge between nodes $v_i$ and $v_j$. Graph Laplacian is defined as $L = D - A$, along with a normalized version $\tilde{L} = I - D^{-1/2} A D^{-1/2}$, where $I$ is the identity matrix and $D$ is a diagonal degree matrix with each diagonal element $D_{ii} = \sum_{i=1}^{N} A_{ij}$. It is known that $\tilde{L}$ is a symmetric positive semidefinite matrix that can be decomposed as $\tilde{L} = U \Lambda U^T$, where $\Lambda = diag\{\lambda_0, \ldots, \lambda_{N-1}\}$ is a diagonal eigenvalue matrix with $0 = \lambda_0 \leq \lambda_1 \leq \ldots \leq \lambda_{N-1} \leq 2$, and $U$ is a unitary matrix consisting of eigenvectors.

For multi-class node classification tasks, nodes in $\mathcal{G}$ are divided into $K$ classes $\{\mathcal{C}_0, \ldots, \mathcal{C}_{K-1}\}$. Each node $v_i$ is attached with an $F$ dimension feature and a one-hot class label. Let $X \in \mathbb{R}^{N \times F}$ be the input feature matrix and $Y \in \mathbb{R}^{N \times K} = (\boldsymbol{y}_0, \ldots, \boldsymbol{y}_{K-1})$ be the label matrix, where $\boldsymbol{y}_i$ is the indicator vector of class $\mathcal{C}_i$. Let $\mathcal{R} = Y^\top Y$ and the size of class $\mathcal{C}_k$ be $\mathcal{R}_k$.

**Graph filtering.** The graph filtering operation on graph signal $X$ is defined as $Z = \sigma(U g(\Lambda) U^T X)$, where $g(\Lambda)$ is the so-called graph filter, and $\sigma$ is the normalization function. Directly learning $g(\Lambda)$ requires eigendecomposition (EVD) of time complexity $O(N^3)$. Recent studies suggest using polynomials to approximate $g(\Lambda)$ instead, which is:

$$U g(\Lambda) U^T X \approx U \left( \sum_{i=0}^{K-1} w_k \Lambda^k \right) U^\top X = \sum_{i=0}^{K-1} w_k \tilde{L}^k X,$$

where $\{w_k\}$ are polynomial coefficients. We can also denote a $K$-order polynomial graph filter as a filter function $g(\lambda) = \sum_{k=0}^{K} w_k \lambda^k$ that maps eigenvalue $\lambda \in [0, 2]$ to $g(\lambda)$.

**Homophily.** Homophily reflects nodes' preferences for choosing neighbors. For a graph of strong homophily, nodes show a tendency to form connections with nodes of the same labels. The ratio of homophily $h$ measures the level of overall homophily in a graph. Several homophily metrics have been proposed with different focuses [20, 22]. We adopt edge homophily following [37], defined by

$$h = \frac{|\{(u, v) : (u, v) \in \mathcal{E} \wedge y_u = y_v\}|}{|\mathcal{E}|}. \tag{1}$$

By definition, $h \in [0, 1]$ is the fraction of intra-class edges in the graph. The closer $h$ is to 1, the more homophilic a graph is.

# 3 Our Proposed Method: EvenNet

In this section, we first introduce our motivation and the methodology of EvenNet. We then explain how EvenNet enhances the robustness of spectral GNNs from the perspective of both spatial and spectral domains.

## 3.1 Motivations

Reconsider the toy example in Figure 1. Relationships between nodes are opposite on homophilic and heterophilic graphs, being straightforward but erratic under changes in the graph structure. Unconstrained spectral GNNs tend to overuse such unstable relationships and fail to generalize across homophily. In contrast, a robust model should rely on more general topological information beyond homophily.

Balance theory [6], which arose from signed networks, offers a good perspective: "The enemy of my enemy is my friend, and the friend of my friend is also my friend." Balance theory always holds as a more general law, regardless of how structural information is revealed on the graph. As a result, we can obtain a more robust spectral GNN under homophily change by incorporating balance theory into graph filter design.

## 3.2 EvenNet

Denote propagation matrix as $P = I - \tilde{L} = D^{-\frac{1}{2}} A D^{-\frac{1}{2}}$. A $K$-order polynomial graph filer is defined as $g(\tilde{L}) = \sum_{k=0}^{K} w_k \tilde{L}^k$, where $w_k, k = 0, \ldots, K$ are learnable parameters. We can rewrite the filter as $g(\tilde{L}) = \sum_{k=0}^{K} w_k (I - \tilde{L})^k = \sum_{k=0}^{K} w_k P^k$ since $w_k$ is learnable. Then, we discard the monomials in $g(\tilde{L})$ containing odd-order $P$, obtaining:

$$g_{\text{even}}(\tilde{L}) = \sum_{k=0}^{\lfloor K/2 \rfloor} w_k (I - \tilde{L})^{2k} = \sum_{k=0}^{\lfloor K/2 \rfloor} w_k P^{2k}. \tag{2}$$

In practice, we decouple the transformation of input features and graph filtering process following [9, 18]. Our model then takes the simple form:

$$Z = f(X, P) = \left( \sum_{k=0}^{\lfloor K/2 \rfloor} w_k P^{2k} t(X) \right), \tag{3}$$

where $t$ is an input transformation function (e.g. MLP), and $Z$ is the output node representation that can be fed into a softmax activation function for node classification tasks.

From the perspective of the spectral domain, $g_{\text{even}}$ keeps both low and high frequencies components and suppresses medium-frequency components, which is a band-reject filter with the filter function symmetric about $\lambda = 1$. We provide a theoretical analysis to demonstrate further the advantages of $g_{\text{even}}$ in Section 3.3 and 3.4.

## 3.3 Analysis from the Spatial Domain

Recently, Chen et al. [8] analyzed the performance of graph filters under certain homophily. They concluded that graph filters operate as a potential reconstruction mechanism of the graph structure. A graph filter $g(\tilde{L})$ achieves **better** performance in a binary node classification task when the homophily of the transformed graph is **high**. The transformed homophily can therefore be seen as an indicator of the performance of graph filters on specific tasks. We now provide the well-defined transformed homophily adopted from [8].

**Definition 1.** *(k-step interaction probability) For propagation matrix $P = D^{-\frac{1}{2}} A D^{-\frac{1}{2}}$, the $k$-step interaction probability matrix is*

$$\tilde{\Pi}^k = \mathcal{R}^{-\frac{1}{2}} Y^\top P^k Y \mathcal{R}^{-\frac{1}{2}}.$$

**Definition 2.** *(k-homophily degree) For a graph $\mathcal{G}$ with the $k$-step interaction probability $\tilde{\Pi}^k$, its $k$-homophily degree $\mathcal{H}_k(\tilde{\Pi})$ is defined as*

$$\mathcal{H}_k(\tilde{\Pi}) = \frac{1}{N} \sum_{l=0}^{K-1} \left( \mathcal{R}_l \tilde{\Pi}_{ll}^k - \sum_{m \neq l} \sqrt{\mathcal{R}_m \mathcal{R}_l} \tilde{\Pi}_{lm}^k \right).$$

*The transformed 1-homophily degree with filter $g(\tilde{L})$ is $\mathcal{H}_1(g(I - \tilde{\Pi}))$.*

By definition, the $k$-homophily degree reflects the average possibility of deriving a node's label from its $k$-hop neighbors. In Theorem 1, we show that even-order filters achieve more robust performance under homophily change by enjoying a lower variance of transformed homophily degree without losing average performance. Detailed proof is provided in the appendix, including discussions about multi-class cases.

**Theorem 1.** *In a binary node classification task, assume the edge homophily $h \in [0, 1]$ is a random variable that belongs to a uniform distribution. An even-order graph filter achieves no less $\mathbb{E}_{\mathcal{H}} \left[ \mathcal{H}_1 \left( g(I - \tilde{\Pi}) \right) \right]$ with lower variation than the full-order version.*

## 3.4 Analysis from Spectral Domain

Similar to Section 3.3, we proposed Spectral Regression Loss (SRL) as an evaluation metric of graph filters in the spectral domain. In a binary node classification task, suppose the dimension of inputs $F = 1$. Denote the difference of labels as $\Delta \boldsymbol{y} = \boldsymbol{y}_0 - \boldsymbol{y}_1$. A graph filtering operation is defined as $Z = \sigma(U g(\Lambda) U^T X)$. Desirable filtering produces distinguishable node representations correlated to $\Delta y$ to identify node labels. Let $\boldsymbol{\alpha} = U^\top \Delta y$ and $\boldsymbol{\beta} = U^\top X$. The classification task in the spectral domain is then a regression problem in the form of $\sigma(\boldsymbol{\alpha}) = \sigma(g(\Lambda)\boldsymbol{\beta})$.

We adopt Mean Squared Error (MSE) as the objective function of the regression problem and vector normalization as $\sigma$. Then SRL is defined as follows:

**Definition 3.** *(Spectral regression loss.) Denote $\boldsymbol{\alpha} = (\alpha_0, \ldots, \alpha_{N-1})^\top, \boldsymbol{\beta} = (\beta_0, \ldots, \beta_{N-1})^\top$. In a binary node classification task, Spectral Regression Loss (SRL) of filter $g(\Lambda)$ on graph $\mathcal{G}$ is:*

$$L(\mathcal{G}) = \sum_{i=0}^{N-1} \left( \frac{\alpha_i}{\sqrt{N}} - \frac{g(\lambda_i)\beta_i}{\sqrt{\sum_{j=0}^{N-1} g(\lambda_j^2)\beta_j^2}} \right)^2 \tag{4}$$

$$= 2 - \frac{2}{\sqrt{N}} \sum_{i=0}^{N-1} \frac{\alpha_i g(\lambda_i)\beta_i}{\sqrt{\sum_{j=0}^{N-1} g(\lambda_j^2)\beta_j^2}}. \tag{5}$$

*The constant $\sqrt{N}$ comes from the fact $\sum_{i=0}^{N-1} \alpha_i^2 = N$. A detailed illustration is included in the appendix. A graph filter that achieves **lower** SRL is of **higher** performance in the task.*

**Filters that Minimize SRL.** Suppose $\alpha_i = w\beta_i + \epsilon$, where $w > 0$ reflects the correlation between labels and features in the spectral domain and $\epsilon$ is the noise term. If $\epsilon$ is close to 0, indicating features are free of noise and highly predictive, an all-pass filter (for example, MLP) with $g(\lambda_i) = 1$ already minimizes SRL. If the noise becomes dominant, SRL approximately equals to $\sum_{i=0}^{N-1}(\frac{\alpha_i}{\sqrt{N}} - \frac{g(\lambda_i)}{\sqrt{\sum_j g(\lambda_j)^2}})^2$. In this noise-dominant case, an ideal filter is linearly correlated to $\boldsymbol{\alpha}$ and structure-based to achieve a lower SRL. Most real-world situations lie between these two opposite settings. As a result, the shape of an ideal graph filter lies between an all-pass filter and an $\boldsymbol{\alpha}$-dependent filter.

From the discussion above, we have shown that the performance of graph filters is related to the correlation between $g(\lambda)$ and $\boldsymbol{\alpha}$. By connecting $\boldsymbol{\alpha}$ and $h$ in Theorem 2, we establish the relationship between graph homophily and the performance of graph filters.

**Theorem 2.** *For a binary node classification task on a $k$-regular graph $\mathcal{G}$, let $h$ be edge homophily and $\lambda_i$ be the $i$-th smallest eigenvalue of $\tilde{L}$, then*

$$1 - h = \frac{\sum_{i=0}^{N-1} \alpha_i^2 \lambda_i}{2 \sum_i \lambda_i} \tag{6}$$

*The above equation can be extended to general graphs by replacing the normalized Laplacian $\tilde{L}$ with the unnormalized $L$.*

Notice that $\sum_{i=0}^{N-1} \alpha_i^2 \lambda_i$ is a convex combination of non-decreasing $\{\lambda_i\}$ with weights $\alpha_i^2$. On a homophilic graph where $h$ is close to 1, the right-hand side of Equation 6 is close to 0, implying larger weights for smaller $\lambda_i$. A low-pass filter that suppresses high-frequency components is more correlated with such $\boldsymbol{\alpha}$ and therefore achieves lower SRL. From previous works, we have known that low-pass filters hold superiority on homophilic graphs, which is consistent with our analysis.

In the case of generalization, the distribution of $\{\alpha_i\}$ is not fixed. A graph filter that minimizes the SRL on training graphs could achieve poor results on a test graph of different homophily. Remember that vanilla GCN could be worse than MLP on many heterophilic graphs. The same conclusion can be applied to learnable filters without any constraints, as they only tried to minimize the SRL of training graphs. In Theorem 3, we prove that even-order design helps spectral GNNs better generalize between homophilic and heterophilic graphs as a practical constraint to current filters.

**Theorem 3.** *Suppose $\lambda_{N-1} = 2$ for a homophilic graph $\mathcal{G}_1$ with non-increasing $\{\alpha_i\}$, and a heterophilic graph $\mathcal{G}_2$ with non-decreasing $\{\alpha_i\}$. Then an even-order filter $g_{even}$ achieves a lower SRL gap $|L(\mathcal{G}_1) - L(\mathcal{G}_2)|$ than full-order filters when trained on one of the graphs and test on the other.*

Theorem 3 reveals a trade-off in filter design between fitting the training graph and generalizing across graphs of different homophily. While naive low-pass filters and high-pass filters work better on graphs with certain homophily, EvenNet tolerates imperfect filter learning and becomes more robust under homophily changes. A specific example on ring graphs is given in the following corollary. We see that EvenNet intrinsically satisfies the necessary condition for perfect generalization.

**Corollary 1.** *Consider two ring graphs $\mathcal{G}_1$ and $\mathcal{G}_2$ of $2n$ nodes, $n \in \mathbb{N}^+$. Suppose $h(\mathcal{G}_1) = 0$ and $h(\mathcal{G}_2) = 1$. Assume the spectrum of input difference $\boldsymbol{\beta} = c\mathbf{1}$, where $c > 0$ is a constant. Then the necessary condition for a graph filter $g(\lambda)$ to achieve $L(\mathcal{G}_1) = L(\mathcal{G}_2)$ is $g(0) = g(2)$.*

### 3.5 Complexity

Denote $N$ the number of nodes, $d$ the size of hidden channels (we assume it is of the same order as the size of input features), $|E|$ the number of edges, $L$ the number of MLP layers used in feature transformation and $K$ the order of the propagation layer.

Compared with structural learning methods which usually have a space complexity of $O(N^2)$, EvenNet takes up $O(|E|)$ space complexity, as it only needs to store the input sparse adjacency matrix during training. For the time complexity of EvenNet, the transform process has a time complexity of $O(Nd^2L)$, and the propagation process has a complexity $O(Kd|E|)$ during each forward pass.

In practice, H2GCN [37] and ProGNN [16] require $O(N^2)$ space complexity and are thus not scalable to large graphs. GCNII [7] achieves its best performance with multiple stacked layers which is slow to train. FAGCN [4], GAT [26] and GNNGuard [34] with attention calculations are also inefficient during training. Notice that the space and time complexity of EvenNet are both linear to $N$ and $|E|$, which is highly efficient.

We report the computational time and an experiment on a larger dataset ogbn-arxiv [15] in the appendix to further verify the efficiency of EvenNet.

## 4 Related Work

**Spectral GNNs.** GNNs have become prevalent in graph representation learning tasks. Among them, Spectral GNNs focus on designing graph filters with filter functions that operate on eigenvalues of graph Laplacian [5]. Graph filters could be fixed [17, 18, 28] or approximated with polynomials. ChebNet [10] adopts Chebyshev polynomials to realize faster localized spectral convolution. ARMA [3] achieves a more flexible filter approximation with Auto-Regressive Moving Average filters. GPRGNN [9] connects graph filtering with graph diffusion and learns coefficients of polynomial filters directly. BernNet [14] utilizes Bernstein approximation to learn arbitrary filtering functions. Although learnable graph filters perform well on heterophilic graphs, they have difficulties generalizing if a homophily gap exists between training and test graphs.

**GNNs for Heterophily.** Previous works pointed out the weakness of vanilla GCN on graphs with heterophily. Recently, various GNNs have been proposed to tackle this problem. Geom-GCN [22] uses a novel neighborhood aggregation scheme to capture long-distance information. Zhu et al. [37] introduces several designs that are helpful for GNNs to learn representations beyond homophily. FAGCN [4] adaptively combines signals of different frequencies in message passing via a self-gating mechanism. While these methods can handle heterophilic graphs, they are not guaranteed to generalize across graphs of different homophily.

**Robust GNNs.** In the field of designing robust GNNs, existing methods can be divided into two main categories: **1) Models utilizing new graph structures.** GNN-Jaccard [29] and GNN-SVD [11] preprocess the input graph before applying vanilla GCN. ProGNN [16] jointly learns a better graph structure and a robust model. **2) Attention-based models.** RGCN [35] uses variance-based attention to evaluate the credibility of nodes' neighbors. GNNGuard [34] adopts neighbor importance estimation, aligning higher scores to trustworthy neighbors. TWIRLS [32] applies an attention mechanism inspired by classical iterative methods PGD and IRLS. These methods are effective against structural attacks. However, the learned graph structure cannot be applied to inductive learning settings and requires additional memory. At the same time, attention-based models are limited in the spatial domain and need high computational costs. On the contrary, EvenNet improves the robustness of spectral GNNs without introducing additional computational costs.

## 5 Experiment

We conduct three experiments to test the ability of EvenNet in (1) generalizing across homophily on synthetic datasets, (2) defending against non-targeted structural attacks, and (3) supervised node classification on real-world datasets.

### 5.1 Baselines

We compare our EvenNet with the following methods. (1) Method only using node features: A 2-layer MLP. (2) Methods achieving promising results on homophilic graphs: GCN [17], GAT [26], GCNII [7]. (3) Methods handling heterophilic settings: H2GCN [37], FAGCN [4], GPRGNN [9]. We also include five advanced defense models in the experiment about adversarial attacks, including RobustGCN [35], GNN-SVD [11], GNN-Jaccard [29], GNNGuard [34], and ProGNN [16]. We implement the above models with the help of PyTorch Geometric [12] and DeepRobust libraries [19].

### 5.2 Evaluation on synthetic datasets

**Datasets.** In the first experiment testing generalization ability, we use cSBM model to generate graphs with arbitrary homophily levels following [9]. Specifically, we divide nodes into two classes of equal size. Each node is attached with a feature vector randomly sampled from a class-specific Gaussian distribution. The homophily level of a graph is controlled by parameter $\phi \in [-1, 1]$. A larger $|\phi|$ indicates that the generated graph provides stronger topological information, while $\phi = 0$ means only node features are helpful for prediction. Note that if $\phi > 0$, the graph is more homophilic and vice versa.

**Settings.** We set up node classification tasks in the inductive setting. We generate three graphs of the same size for each sub-experiment, one graph each for training, validation, and testing. Graphs for validation and testing share the same $\phi_{test}$, while training graphs either take $\phi_{train} = \phi_{test}$ or $\phi_{train} = -\phi_{test}$. If $\phi_{train} = -\phi_{test}$, the training and test graphs are of opposite homophily but provide the same amount of topological information. A model manages to generalize across homophily when it realizes high prediction accuracy in both scenarios. In practice, we choose $(\phi_{train}, \phi_{test}) \in \{(\pm 0.5, \pm 0.5), (\pm 0.75, \pm 0.75)\}$.

**Results.** The results are presented in Table 1. When $\phi_{train} = \phi_{test}$, GPRGNN achieves the highest predictive accuracy as it best fits the desired graph filter. However, when $\phi_{train} = -\phi_{test}$, all methods except EvenNet suffer from a huge performance drop. Vanilla GCN, which corresponds to a low-pass filter, achieves desirable performance only when the test graph is homophilic. GPRGNN overfits training graphs most, resulting in more severe performance degradation on test graphs of opposite homophily. EvenNet is the only method that achieves more than 75% accuracy on all datasets among all the models, which is robust in generalization across homophily.

Table 1: Average node classification accuracy(%) and absolute performance gap(%) between experiments of the same $\phi_{train}$ over ten repeated experiments on synthetic cSBM datasets. The best result is highlighted by **bold** font, the second best result is underlined.

| $\phi_{train}$ | | 0.75 | | | 0.50 | | | -0.50 | | | -0.75 | |
| $\phi_{test}$ | 0.75 | -0.75 | gap(↓) | 0.50 | -0.50 | gap(↓) | -0.50 | 0.50 | gap(↓) | -0.75 | 0.75 | gap(↓) |
|---|---|---|---|---|---|---|---|---|---|---|---|---|
| MLP | 57.92 | 57.24 | **0.68** | 63.65 | 64.26 | 0.61 | 63.28 | 63.83 | **0.55** | 56.92 | 59.24 | 2.32 |
| GCN | 75.24 | 60.31 | 15.11 | 78.98 | 63.21 | 15.77 | 63.27 | 76.67 | 13.40 | 60.48 | 77.88 | 17.40 |
| GAT | 74.15 | 60.55 | 13.60 | 75.64 | 61.96 | 13.68 | 64.43 | 71.02 | 6.59 | 63.19 | 71.61 | 8.42 |
| GCNII | 83.12 | 54.30 | 28.82 | 78.07 | 58.43 | 19.64 | 72.32 | 67.68 | 4.64 | 65.93 | 62.92 | 3.01 |
| H2GCN | 76.41 | 54.81 | 21.60 | 78.86 | 58.89 | 19.97 | 78.43 | 59.77 | 18.66 | 76.29 | 55.92 | 20.37 |
| FAGCN | 81.29 | 60.44 | 20.85 | 78.73 | 60.28 | 18.45 | 79.45 | 60.62 | 18.83 | 85.78 | 57.34 | 28.44 |
| GPRGNN | **95.93** | 53.52 | 42.41 | **84.42** | 56.16 | 28.26 | **84.18** | 63.76 | 20.42 | **95.99** | 66.49 | 29.52 |
| EvenNet | 95.29 | **94.59** | 0.70 | 82.37 | **82.57** | **0.20** | 81.99 | **79.81** | 2.18 | 94.79 | **96.25** | **1.46** |

### 5.3 Performance under non-targeted structural adversarial attacks

**Datasets.** For adversarial attacks, we use four public graphs, Cora, Citeseer, PubMed [24, 33] and ACM [30] available in DeepRobust Library [19]. We use the same preprocessing method and splits as [38], where the node set is split into 10% for training, 10% for validation, and 80% for testing, and the largest connected component of each graph for attacks are selected.

We include the experiment against non-targeted attacks on heterophilic datasets in the appendix as well, in which we use the same preprocessing methods and dense splits following [9]

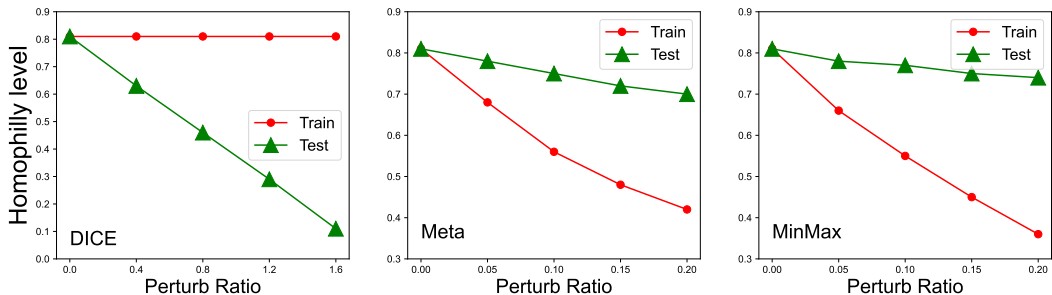

Figure 2: Homophily level of training graphs and test graphs on Cora after DICE attack, Metattack, and MinMax attack. All attacks result in a homophily gap between training and test graphs.

.

**Attack methods.** Graph structural attacks can be categorized into poison attacks and evasion attacks. In poison attacks, attack models are trained to lower the performance of a surrogate GNN model. The training graph and the test graph are both allowed to be perturbed but only with a limited amount of modifications, which are referred to as perturb ratios. Evasion attacks only happen during inference, meaning GNNs are trained on clean graphs. In our study, we include two poison attacks, Metattack (Meta) [38] and MinMax attack [31] with GCN the surrogate model, and an evasion variant of DICE attack [27]. Notice that we mainly focus on modification attacks, which are strictly structural attacks. A discussion of GNNs under graph injection attacks is included in the appendix.

For poison attacks, we use the same setting in [34] and set the perturb ratio for poison attacks to be 20%. For the evasion DICE, we randomly remove intra-class edges and add inter-class edges on the test graph while keeping the graph structure between labeled nodes unchanged. We set the perturb ratio of DICE attack in $\{0.4, 0.8, 1.2, 1.6\}$. From Figure 2, it can be seen that all attacks result in homophily gaps between the training graphs and the test graphs. Besides the 1-hop homophily gap, in Table 2, we present the change in two-hop homophily for learnable attacks with perturb ratio of 0.2. As can be seen, the two-hop homophily gap is relatively smaller than the one-hop homophily gap, which is in accordance with our analysis that homophily between even-hop neighbors is more robust.

Table 2: Homophily gaps between training and test graphs after Meta/MinMax attacks with 20% the perturb ratio.

| Homophily \dataset | Meta-Cora | Meta-Citeseer | Meta-ACM | MinMax-Cora | MinMax-Citeseer | MinMax-ACM |
|---|---|---|---|---|---|---|
| 1-hop Train | 0.42 | 0.4 | 0.49 | 0.36 | 0.38 | 0.49 |
| 1-hop Test | 0.7 | 0.65 | 0.72 | 0.74 | 0.69 | 0.72 |
| **1-hop Gap** | **0.28** | **0.25** | **0.23** | **0.38** | **0.31** | **0.23** |
| 2-hop Train | 0.52 | 0.55 | 0.54 | 0.37 | 0.40 | 0.36 |
| 2-hop Test | 0.65 | 0.66 | 0.61 | 0.69 | 0.68 | 0.56 |
| **2-hop Gap** | **0.13** | **0.11** | **0.07** | **0.32** | **0.28** | **0.20** |

**Results.** Defense results are presented in Table 3 and Figure 3. For the DICE attack, the performance of all methods significantly decreases along with the increase of the homophily gap except EvenNet. Interestingly, when the homophily gap is enormous, EvenNet enjoys a performance rebound, consistent with our topological information theory (strong homo. and strong hetero. are both helpful for prediction). For poison attacks, EvenNet achieves SOTA compared with advanced defense models. Unlike spatial defense models, EvenNet is free of introducing extra time or space complexity.

### 5.4 Performance on real-world graph datasets

We evaluate EvenNet on real-world datasets to examine the performance of EvenNet on clean graphs. Besides the datasets used in Section 5.3, we additionally include four public heterophilic datasets: Actor, Cornell, Squirrel, and Texas [22, 23, 25]. The statistics of real-world Datasets are included in Table 4. In the node classification task, we transform heterophilic datasets into undirected ones following [9].

Table 3: Average node classification accuracy (%) against non-targeted poison attacks Metattack and MinMax attack with perturb ratio 20% over 5 different splits. The best result is highlighted by **bold** font, the second best result is underlined.

| Dataset | Meta-cora | Meta-citeseer | Meta-acm | MM-cora | MM-citeseer | MM-acm |
|---|---|---|---|---|---|---|
| MLP | 58.60 | 62.93 | 85.74 | 59.81 | 63.72 | 85.66 |
| GCN | 63.76 | 61.98 | 68.29 | 69.21 | 68.02 | 69.37 |
| GAT | 66.51 | 63.66 | 68.50 | 69.50 | 67.04 | 69.26 |
| GCNII | 66.57 | 64.23 | 78.53 | 73.01 | 72.26 | 82.90 |
| H2GCN | 71.62 | 67.26 | 83.75 | 66.76 | 69.66 | 84.84 |
| FAGCN | 72.14 | 66.59 | 85.93 | 64.90 | 66.33 | 81.49 |
| GPRGNN | 76.27 | 69.63 | 88.79 | 77.18 | 72.81 | 88.24 |
| RobustGCN | 60.38 | 60.44 | 62.29 | 68.53 | 63.16 | 61.60 |
| GNN-SVD | 64.83 | 64.98 | 84.55 | 66.33 | 64.97 | 81.08 |
| GNN-Jaccard | 68.30 | 63.40 | 67.81 | 72.98 | 68.43 | 69.03 |
| GNNGuard | 75.98 | 68.57 | 62.19 | 73.23 | 66.14 | 66.15 |
| ProGNN | 75.25 | 68.15 | 83.99 | 77.91 | 72.26 | 73.51 |
| EvenNet | **77.74** | **71.03** | **89.78** | **78.40** | **73.51** | **89.80** |

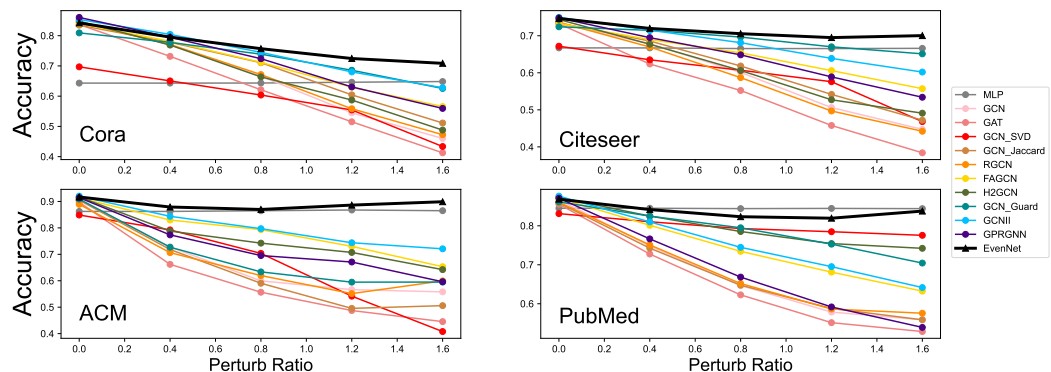

Figure 3: DICE attack on four homophilic datasets. EvenNet is marked with "△".

For all datasets, we adopt dense splits the same as [22] to perform full-supervised node classification tasks, where the node set is split into 60% for training, 20% for validation, and 20% for testing.

Table 4: Statistics of real-world datasets.

| | Cora | Citeseer | PubMed | ACM | Chameleon | Squirrel | Cornell | Texas | Actor |
|---|---|---|---|---|---|---|---|---|---|
| Nodes | 2,708 | 3,327 | 19,717 | 3,025 | 2,277 | 5,201 | 183 | 183 | 7,600 |
| Edges | 5,278 | 4,552 | 44,324 | 13,128 | 31,371 | 198,353 | 277 | 279 | 26,659 |
| Features | 1,433 | 3,703 | 500 | 1,870 | 2,325 | 2,089 | 1,703 | 1,703 | 932 |
| Classes | 7 | 6 | 3 | 3 | 5 | 5 | 5 | 5 | 5 |
| Homophily Level | 0.81 | 0.74 | 0.80 | 0.82 | 0.23 | 0.22 | 0.30 | 0.09 | 0.22 |

The results are shown in Table 5. While EvenNet sacrifices its performance for robustness, it is still competitive on most datasets.

## 5.5 Ablation study

To analyze the effect of introducing odd-order components into graph filters, we develop a regularized variant of EvenNet named EvenReg. EvenReg adopts a full-order learnable graph filter, with the coefficients of odd-order monomials being punished as a regularization term. The training loss of EvenReg then takes the form: $\mathcal{L} = \mathcal{L}_{pred} + \eta \sum_{k=0}^{\lfloor K/2 \rfloor} |w_{2k+1}|$, where $\mathcal{L}_{pred}$ is the classification loss and $\eta$ is a hyper-parameter controlling the degree of regularization.

Table 5: Average node classification accuracy(%) on real-world benchmark datasets over 10 different splits. The best result is highlighted by **bold** font, the second best result is underlined.

| Model | Cora | Cite. | Pubm. | Cham. | Texas | Corn. | Squi. | Actor |
|-------|------|-------|-------|-------|-------|-------|-------|-------|
| MLP | 74.88 | 74.82 | 85.58 | 46.65 | 89.50 | 90.17 | 32.33 | 41.30 |
| GCN | 87.19 | 80.87 | 87.51 | 63.28 | 80.66 | 74.09 | 46.42 | 34.21 |
| GAT | 88.21 | 81.36 | 89.42 | 64.02 | 81.63 | 81.97 | 47.87 | 36.21 |
| GCNII | 87.91 | **82.13** | 86.41 | 50.76 | 86.23 | 89.83 | 36.35 | **41.68** |
| FAGCN | **88.83** | 80.35 | 89.34 | 56.67 | 89.18 | 90.16 | 39.10 | 41.18 |
| H2GCN | 87.59 | 79.69 | 88.68 | 55.88 | 88.52 | 85.57 | 34.45 | 39.62 |
| GPRGNN | 88.34 | 80.16 | **90.08** | **67.13** | 93.44 | **92.45** | **51.93** | 41.62 |
| EvenNet | 87.25 | 78.65 | 89.52 | 66.13 | **93.77** | 92.13 | 49.80 | 40.48 |

We set $\eta = 0.05$ and repeat experiments in Section 5.2. The results are presented in Table 6. The performance of EvenReg lies between full-order GPRGNN and EvenNet, indicating the introduced odd orders impede spectral GNNs to generalize across homophily.

Table 6: Average node classification accuracy(%) of EvenReg over 10 repeated experiments on synthetic cSBM datasets.

| $\phi_{train}$ | 0.75 | | 0.50 | | -0.50 | | -0.75 | |
|----------------|------|------|------|------|-------|------|-------|------|
| $\phi_{test}$ | 0.75 | -0.75 | 0.50 | -0.50 | -0.50 | 0.50 | -0.75 | 0.75 |
| GPRGNN | **95.93** | 53.52 | **84.42** | 56.16 | **84.18** | 63.76 | **95.99** | 66.49 |
| EvenNet | 95.29 | **94.59** | 82.37 | **82.57** | 81.99 | **79.81** | 94.79 | **96.25** |
| EvenReg | 95.44 | 93.90 | 84.05 | 78.06 | 83.72 | 75.33 | 95.40 | 95.73 |

# 6 Conclusion

In this study, we investigate the ability of current GNNs to generalize across homophily. We observe that all existing methods experience severe performance degradation if a large homophily gap exists between training and test graphs. To overcome this difficulty, we proposed EvenNet, a simple yet effective spectral GNN which is robust under homophily change of graphs. We provide a detailed theoretical analysis to illustrate the advantages of EvenNet in generalization between graphs with homophily gaps. We conduct experiments on both synthetic and real-world datasets. The empirical results verify the superiority of EvenNet in inductive learning across homophily and defense under non-targeted structural attacks by sacrificing only a tiny amount of predictive accuracy on clean graphs.

# 7 Acknowledgement

This research was supported in part by the major key project of PCL (PCL2021A12), by National Natural Science Foundation of China (No. 61972401, No. 61932001, No. 61832017), by Beijing Natural Science Foundation (No. 4222028), by Beijing Outstanding Young Scientist Program No. BJJWZYJH012019100020098, by Alibaba Group through Alibaba Innovative Research Program, by CCF-Baidu Open Fund (NO.2021PP15002000) and by Huawei-Renmin University joint program on Information Retrieval. We also wish to acknowledge the support provided by Engineering Research Center of Next-Generation Intelligent Search and Recommendation, Ministry of Education. Additionally, we acknowledge the support from Intelligent Social Governance Interdisciplinary Platform, Major Innovation & Planning Interdisciplinary Platform for the "Double-First Class" Initiative, Public Policy and Decision-making Research Lab, Public Computing Cloud, Renmin University of China.

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
