# A  Additional proofs

## A.1  Proof of Theorem 1

Before proofing Theorem 1, We first demonstrate the superiority of even-hop neighbors over odd-hop neighbors from the perspective of random walks.

In a binary node classification task, denote the probability of a random walk of length $k$ that starts and ends with nodes of the same label as $p_k, k > 0$. Suppose the edge homophily level $h$ is a random variable that belongs to a uniform distribution in $[0, 1]$ and $p_1 = h$, then:

**Lemma 1.** *If $k$ is odd, $\mathbb{E}_h[p_k] = \frac{1}{2}$. If $k$ is even, $\mathbb{E}_h[p_k] \geq \frac{1}{2}$.*

*Proof.* If $k = 1$, $p_1 = h$, $\mathbb{E}_h[p_1] = \mathbb{E}_h[h] = \frac{1}{2}$. If $k = 2$, $p_2 = h^2 + (1-h)^2 = 2(h - \frac{1}{2})^2 + \frac{1}{2} \geq \frac{1}{2}$, $\mathbb{E}_h[p_2] = \int_0^1 p_2 dh \geq \frac{1}{2}$.

For $k > 2$,

$$
\begin{aligned}
p_k &= p_{k-2}\left((1-h)^2 + h^2\right) + (1 - p_{k-2})\left(2h(1-h)\right) \\
&= 4p_{k-2}h^2 - 4p_{k-2}h + p_{k-2} - 2h^2 + 2h \\
\mathbb{E}_h[p_k] &= \mathbb{E}_h\left[\mathbb{E}_h[p_k \mid p_{k-2}]\right] \\
&= \mathbb{E}_h\left[\int_0^1 p_k dh\right] \\
&= \mathbb{E}_h\left[\frac{1}{3}p_{k-2} + \frac{1}{3}\right] \\
&= \frac{1}{3}\left(\mathbb{E}_h[p_{k-2} + 1]\right)
\end{aligned}
$$

That is, for $\mathbb{E}_h[p_{k-2}] = \frac{1}{2}$, $\mathbb{E}_h[p_k] = \frac{1}{2}$; for $\mathbb{E}_h[p_{k-2}] \geq \frac{1}{2}$, $\mathbb{E}_h[p_k] \geq \frac{1}{2}$. Therefore, Lemma 1 is proved. □

**Multi-class Cases.** We now provide a brief discussion of the superiority of even-hop neighbors in multi-class node classification tasks following [14].

**Definition 1.** *The matrix $Q \in \mathbb{R}^{K \times K}$ is an independent between-class random walk matrix if it holds the following properties:*

- *$Q$ is a random walk matrix.*

- *$\forall i \neq j, Q_{ii} = Q_{jj}$.*

- *$\forall i \neq j, m \neq n, Q_{ij} = Q_{mn}$.*

Suppose there are $K$ classes of nodes in the graph, where the number of nodes of each class is the same, and node labels are assigned independently. Denote $h$ as the edge homophily level, the 1-step between-class random walk matrix $P$ is in the form of:

$$
P = \begin{bmatrix}
h & \frac{1-h}{K-1} & \cdots & \frac{1-h}{K-1} \\
\frac{1-h}{K-1} & h & \cdots & \frac{1-h}{K-1} \\
\vdots & \vdots & \ddots & \vdots \\
\frac{1-h}{K-1} & \frac{1-h}{K-1} & \cdots & h
\end{bmatrix},
$$

where $P_{ij}$ denotes the probability of a 1-step random walk that start with a node of label $i$ and ends with a node of label $j$. By definition, $P$ is an independent between-class random walk matrix.

**Lemma 2.** *If $M \in \mathbb{R}^{K \times K}$ is an independent between-class random walk matrix, $P' = MP$ is an independent between-class random walk matrix as well.*

*Proof.* It can be verified that $P'$ is still a random walk matrix, and for all $i \neq j, m \neq n$:

$$P'_{ii} = \sum_{m=0}^{K-1} M_{im} P_{mi} = \sum_{m=0}^{K-1} M_{jm} P_{mj} = P'_{jj}$$

$$P'_{ij} = \sum_{k=0}^{K-1} M_{ik} P_{kj} = \sum_{k=0}^{K-1} M_{mk} P_{kn} = P'_{mn}$$

By definition, $P'$ is an independent between-class random walk matrix. □

Denote the $k$-step between-class random walk matrix as $P^k$. From Lemma 2, we can conclude that $P^k$ is an independent between-class random walk matrix for all $k \in \mathbb{N}^+$. In Lemma 3, we illustrate the advantages of even-order propagation by comparing the interaction probability between classes.

**Lemma 3.** *If $k$ is even, the intra-class interaction probability $P_{ii}^k$ is no less than inter-class interaction probability $P_{ij}^k, i \neq j$.*

*Proof.* For $k = 2$:

$$P_{ii}^2 = h^2 + \frac{(1-h)^2}{K-1}$$

$$P_{ij}^2 = \frac{2h(1-h)}{K-1} + \frac{(K-2)(1-h)^2}{(K-1)^2}$$

$$P_{ii}^2 - P_{ij}^2 = \left( h - \frac{1-h}{K-1} \right)^2 \geq 0$$

The inequality is tight when $h = \frac{1-h}{K-1}$.

For $k = 2m, m > 1, m \in \mathbb{N}^+$, $P^m$ is an independent between-class random walk matrix that can be written as:

$$P^m = \begin{bmatrix} h' & \frac{1-h'}{K-1} & \cdots & \frac{1-h'}{K-1} \\ \frac{1-h'}{K-1} & h' & \cdots & \frac{1-h'}{K-1} \\ \vdots & \vdots & \ddots & \vdots \\ \frac{1-h'}{K-1} & \frac{1-h'}{K-1} & \cdots & h' \end{bmatrix},$$

The above proof for $k = 2$ can be generalized to all $h \in [0, 1]$. Therefore, $P_{ii}^m \geq P_{ij}^m$ is satisfied as well. □

Note that Lemma 3 is not satisfied for odd $k$ if $h$ is relatively smaller than $\frac{1-h}{K-1}$.

**Proof of Theorem 1**

*Proof.* According to Definition 1, for a graph $\mathcal{G}$ with the $k$-step interaction probability $\tilde{\Pi}^k$, its $k$-homophily degree $\mathcal{H}_k(\tilde{\Pi})$ is defined as

$$\mathcal{H}_k(\tilde{\Pi}) = \frac{1}{N} \sum_{l=0}^{K-1} \left( \mathcal{R}_l \tilde{\Pi}_{ll}^k - \sum_{m \neq l} \sqrt{\mathcal{R}_m \mathcal{R}_l} \tilde{\Pi}_{lm}^k \right).$$

The transformed 1-homophily degree with filter $g(\tilde{L})$ is $\mathcal{H}_1(g(I - \tilde{\Pi}))$.

Specifically, in a binary node classification problem, the $k$-homophily degree is:

$$\mathcal{H}_k(\tilde{\Pi}) = \frac{1}{N} \left( \mathcal{R}_0 \tilde{\Pi}_{00}^k + \mathcal{R}_1 \tilde{\Pi}_{11}^k - 2\sqrt{\mathcal{R}_0 \mathcal{R}_1} \tilde{\Pi}_{01}^k \right)$$

Denote a $k$-order polynomial graph filter as $g_k(\tilde{L}) = w_0 + w_1 \tilde{L} + \ldots + w_k \tilde{L}^k$,

The transformed 1-homophily degree of $g_k(\tilde{L})$ is:

$$\mathcal{H}_1\left(g_k(I-\tilde{\Pi})\right) = \frac{1}{N}\left(\mathcal{R}_0 g_k\left(I-\tilde{\Pi}\right)_{00} + \mathcal{R}_1 g_k\left(I-\tilde{\Pi}\right)_{11} - 2\sqrt{\mathcal{R}_0\mathcal{R}_1}g_k\left(I-\tilde{\Pi}\right)_{01}\right)$$

$$= \frac{(\sqrt{\mathcal{R}_0}-\sqrt{\mathcal{R}_1})^2}{N}(w_0+\ldots+w_k)$$

$$- \frac{\mathcal{R}_0\tilde{\Pi}_{00}+\mathcal{R}_1\tilde{\Pi}_{11}-2\sqrt{\mathcal{R}_0\mathcal{R}_1}\tilde{\Pi}_{01}}{N}(w_1+2w_2+\ldots+kw_k)$$

$$+ \frac{\mathcal{R}_0\tilde{\Pi}_{00}^2+\mathcal{R}_1\tilde{\Pi}_{11}^2-2\sqrt{\mathcal{R}_0\mathcal{R}_1}\tilde{\Pi}_{01}^2}{N}(w_2+3w_3+\ldots)$$

$$- \ldots$$

$$= c - \mathcal{H}_1(\tilde{\Pi})(w_1+2w_2+\ldots+kw_k) + \mathcal{H}_2(\tilde{\Pi})(w_2+3w_3+\ldots) + \ldots$$

$$= c + \sum_{i=1}^{k}\theta_i\mathcal{H}_i(\tilde{\Pi}),$$

where $c$ is a constant and each $\theta_i$ is a linear sum of $\{w_i\}$ that for arbitrary $x\in\mathbb{R}$:

$$\sum_{i=0}^{k}\theta_i x^i = \sum_{i=0}^{k}w_i(1-x)^i.$$

Following Lemma 1, the average possibility of deriving a node's label from its odd-hop neighbors is $\frac{1}{2}$, which means $\mathbb{E}_h[\mathcal{H}_i(\tilde{\Pi})]=0$ and $\text{Var}_h[\mathcal{H}_i(\tilde{\Pi})]\geq 0$ for odd $i$. By removing the odd-order terms, the transformed 1-homophily degree does not decrease on average but enjoys a lower variation.

Rewrite filter $g_k(\tilde{L})$ as $g_k(\tilde{L})=\theta_0+\theta_1(I-\tilde{L})+\ldots+\theta_k(I-\tilde{L})^k$ and set $\theta_i=0$ for odd $i$, the graph filter is then in the form of :

$$g_k(\tilde{L}) = \sum_{i=0}^{\lfloor k/2\rfloor}\theta_i(I-\tilde{L})^{2i} = \sum_{i=0}^{\lfloor k/2\rfloor}\theta_i P^{2i},$$

which is exactly the graph filter of EvenNet. Therefore, EvenNet has a lower variation on the transformed 1-homophily degree without sacrificing the average performance.

$\square$

## A.2 Proof of Theorem 2

**Lemma 4.** *Given normalized graph Laplacian $\tilde{L}=U\Lambda U^\top$ and label difference as $\Delta\boldsymbol{y}=\boldsymbol{y_0}-\boldsymbol{y_1}$, unnormalized $\alpha=U^\top\Delta\boldsymbol{y}=(\alpha_0,\cdots,\alpha_{N-1})^\top$ satisfies $\sum_{i=0}^{N-1}\alpha_i^2=N$.*

*Proof.* Since $\tilde{L}$ is a real symmetric matrix, $U^T$ can be chosen to be an orthogonal matrix. Denote the element of the $i$-th row and $j$-th column of $U^T$ as $u_{ik}$.

$$\sum_{i=0}^{N-1}\alpha_i^2 = \sum_{i=0}^{N-1}\left(\sum_{j=0}^{N-1}u_{ij}\Delta\boldsymbol{y}(j)\right)^2$$

$$= \sum_{i=0}^{N-1}\sum_{j=0}^{N-1}u_{ij}^2 + 2\sum_{i=0}^{N-1}\left(\sum_{j=0}^{N-1}\sum_{k=0}^{N-1}u_{ij}u_{ik}\Delta\boldsymbol{y}(j)\Delta\boldsymbol{y}(k)\mathbf{1}\{j\neq k\}\right)$$

$$= \sum_{i=0}^{N-1}\sum_{j=0}^{N-1}u_{ij}^2 + 2\sum_{j=0}^{N-1}\sum_{k=0}^{N-1}\left(\Delta\boldsymbol{y}(j)\Delta\boldsymbol{y}(k)\sum_{i=0}^{N-1}u_{ij}u_{ik}\mathbf{1}\{j\neq k\}\right)$$

$$= \sum_{i=0}^{N-1}\sum_{j=0}^{N-1}u_{ij}^2 \qquad\qquad\qquad \text{(Orthogonality)}$$

$$= N$$

$\square$

Lemma 4 provides the relationship between normalized and unnormalized $\alpha$, which is also helpful in defining the SRL loss.

**Lemma 5.** *Given **unnormalized graph Laplacian** $L = U_L \Lambda_L U_L^\top$ and its eigenvalues $\{\lambda_i'\}$, denote the number of edges on the graph is $m$, and label difference as $\Delta \boldsymbol{y} = \boldsymbol{y_0} - \boldsymbol{y_1} \in \mathbb{R}^{N \times 1}$. For **unnormalized** spectrum of label difference on $L$ is $\boldsymbol{\alpha}' = U_L^\top \Delta \boldsymbol{y} = (\alpha_0', \alpha_1', \dots, \alpha_{N-1}')^\top$, then*

$$\sum_{i=0}^{N-1} \lambda_i' = 2m \tag{1}$$

$$1 - h = \frac{\sum_{i=0}^{N-1} (\alpha_i')^2 \lambda_i'}{2 \sum_{j=0}^{N-1} \lambda_j'}$$

*Proof.* Denote the trace of a matrix $M$ as $tr(M)$, the degree of node $v_i$ as $d_i$.

$$\sum_{i=0}^{N-1} \lambda_i' = tr(L) = \sum_{i=0}^{N-1} d_i = 2m$$

The Dirichlet energy of the label difference is defined as:

$$\begin{aligned} E(\Delta \boldsymbol{y}) &= \Delta \boldsymbol{y}^\top L \Delta \boldsymbol{y} \\ &= \sum_{(i,j) \in \mathcal{E}} (\Delta \boldsymbol{y}_i - \Delta \boldsymbol{y}_j)^2 \\ &= 4 \sum_{(i,j) \in \mathcal{E}} \mathbf{1}\{\Delta \boldsymbol{y}_i \neq \Delta \boldsymbol{y}_j\} \\ &= 4(1 - h)m \end{aligned} \tag{2}$$

Using $L = U_L \Lambda_L U_L^T$, the Dirichlet energy of the label difference can also be expressed as:

$$\begin{aligned} E(\Delta \boldsymbol{y}) &= \Delta \boldsymbol{y}^T U_L \Lambda_L U_L^\top \Delta \boldsymbol{y} \\ &= \boldsymbol{\alpha'}^\top \Lambda_L \boldsymbol{\alpha'} \\ &= \sum_{i=0}^{N-1} \lambda_i' (\alpha_i')^2 \end{aligned} \tag{3}$$

By integrating equations 1 2 3, we get:

$$1 - h = \frac{\sum_{i=0}^{N-1} (\alpha_i')^2 \lambda_i'}{2 \sum_{j=0}^{N-1} \lambda_j'}$$

$\square$

**Proof of Theorem 2**

*Proof.* For a $k$-regular graph, denote normalized graph Laplacian as $\tilde{L} = U \Lambda U^\top$, then

$$\begin{aligned} \tilde{L} &= D^{-1/2} L D^{-1/2} \\ &= D^{-1/2} U_L \Lambda_L U_L^\top D^{-1/2} \\ &= \frac{1}{k} U_L \Lambda_L U_L^\top \end{aligned} \tag{4}$$

From equation 4, we get $U = U_L$, $\lambda_i = \lambda_i'/k$. Denote the spectrum of label difference on $\tilde{L}$ as $\boldsymbol{\alpha}$, then $\boldsymbol{\alpha} = \boldsymbol{\alpha}'$. By substituting $\lambda_i'$ with $k\lambda_i$ in Lemma 5, we acquire the equation in Theorem 2.

If $\boldsymbol{\alpha}$ is normalized as in the SRL that satisfies $\sum_{i=0}^{N-1} \alpha_i^2 = 1$, following Lemma 4, Theorem 2 is in the form of:

$$1 - h = \frac{N \sum_{i=0}^{N-1} (\alpha_i)^2 \lambda_i}{2 \sum_{j=0}^{N-1} \lambda_j}$$

$\square$

## A.3 Proof of Theorem 3

*Proof.* Denote filter $g(\lambda)$ as $g(\lambda) = \sum_{i=0}^{K} w_i(1-\lambda)^i$, $g_{even}(\lambda) = \frac{1}{2}(g(\lambda) + g(2-\lambda))$ and $g_{odd} = g(\lambda) - g_{even}(\lambda)$. The filter $g_{even}(\lambda)$ is free of odd order terms. The odd filter $g_{odd}(\lambda)$ can be seen as the gap between the full-order filter and the even-order filter.

$$g_{even}(\lambda) = \sum_{k=0}^{\lfloor K/2 \rfloor} w_k(1-\lambda)^{2k}$$

$$g_{odd}(\lambda) = \sum_{k=0}^{\lfloor K/2 \rfloor} w_k(1-\lambda)^{2k+1}.$$

We now consider the SRL gap of the odd-order filter to illustrate the effect of removing odd-order terms. The regression problem in the spectral domain with normalized $\boldsymbol{\alpha}$: $\boldsymbol{\alpha} = \sigma(g(\Lambda)\boldsymbol{\beta})$, $\sum_{i=0}^{N-1} \alpha_i^2 = 1$. Suppose $\boldsymbol{\alpha}$ and $\boldsymbol{\beta}$ is positive correlated in the form of $\mathbb{E}[\boldsymbol{\alpha}] = w\boldsymbol{\beta}, w > 0$, and $\lambda_{N-1} = 2$ for both $\mathcal{G}_1$ and $\mathcal{G}_2$ (to ensure both graphs can achieve $h = 0$).

The SRL of filter $g_{odd}$ and $g_{even}$ between normalized $\boldsymbol{\alpha}$ and normalized $g(\boldsymbol{\lambda})\boldsymbol{\beta}$ is:

$$\begin{aligned}
L_{odd}(\mathcal{G}) &= 2 - \frac{1}{T_{odd}} \left( \sum_{i=0}^{N} \alpha_i^2 g(\lambda_i)_{odd} \right) \\
&= 2 - \frac{1}{T_{odd}} \left( \sum_{i=0}^{N//2} \alpha_i^2 g(\lambda_i)_{odd} - \sum_{i=N//2}^{N} \alpha_i^2 |g(\lambda_i)_{odd}| \right) \\
&= 2 - \frac{1}{T_{odd}} \left( \sum_{i=0}^{N//2} (\alpha_i^2 - \alpha_{N-i-1}^2) g(\lambda_i)_{odd} \right) = 2 - L_o \quad (5)
\end{aligned}$$

$$L_{even}(\mathcal{G}_1) = 2 - \frac{1}{T_{even}} \left( \sum_{i=0}^{N//2} (\alpha_i^2 + \alpha_{N-i-1}^2) g(\lambda_i)_{even} \right) = 2 - L_e, \quad (6)$$

where $T_{type} = \sqrt{\sum_{i=0}^{N-1} g_{type}(\lambda_i)^2 \alpha_i^2}$.

Compare equations 5 and 6. If $g(\lambda_i)$ is of different monotonicity against $\{\alpha_i\}$, which happens when a trained odd-filter is generalized to graphs of opposite homophily, $L_o$ becomes negative. In contrast, $L_e$ is always positive and benefits from reducing SRL.

Suppose $L_o$ is the approximate SRL gap between $g(\lambda)$ and $g_{even}(\lambda)$. The instability of $L_o$ implies $L_g(\mathcal{G}_{train}) < L_{g_{even}}(\mathcal{G}_{train})$ and $L_g(\mathcal{G}_{test}) > L_{g_{even}}(\mathcal{G}_{test})$, reflecting a larger SRL gap of full-order filters than the even-order filters.

More generally, for the cases where $\lambda_{N-1} < 2$, we can still adopt the idea of discarding odd-order terms. Rewrite $g(\lambda)$ as $g'(\lambda) = \sum_{i=0}^{K} (\lambda_{mid} - \lambda)^i$, where $\lambda_{mid}$ is the median of $\{\lambda_i\}$. By applying the same analysis above, we can see that removing odd-order terms from $g'(\lambda)$ is still beneficial to narrow the SRL gap. $\qquad\square$

## A.4 Proof of Corollary 1

*Proof.* In the case where $h(\mathcal{G}_1) = 0$ and $h(\mathcal{G}_2) = 1$, denote the label difference of $\mathcal{G}_1$ and $\mathcal{G}_2$ as $\Delta\boldsymbol{y}_1$ and $\Delta\boldsymbol{y}_2$, where $\Delta\boldsymbol{y}_1 = (1, -1, 1, -1, \ldots, 1, -1)^\top$, $\Delta\boldsymbol{y}_2 = (1, \ldots, 1)^\top$.

Denote $U = (\boldsymbol{u}_0, \boldsymbol{u}_1, \ldots, \boldsymbol{u}_{N-1})^\top$, where $\boldsymbol{u}_i$ is the $i$-th eigenvector of $U$ and $u_k(n)$ is the $n$-th element of $\boldsymbol{u}_k$.

On ring graphs, denote $a_{kn} = \sin(\pi(k+1)n/N, b_{kn} = \cos(\pi kn/N)$, then the normalized $u_k(n)$ satisfies:

$$
u_k(n) = \begin{cases}
\frac{a_{kn}}{\sqrt{N/2}}, & \text{for odd } k, k < N-1 \\
\frac{b_{kn}}{\sqrt{N/2}}, & \text{for even } k \\
\frac{\cos(\pi n)}{\sqrt{N}}, & \text{for odd } k, k = N-1 \\
\frac{1}{\sqrt{N}} & \text{for even } k, k = 0
\end{cases}
$$

Let the normalized spectrum of $\mathcal{G}_1$ be $\boldsymbol{\delta} = U^T \Delta \boldsymbol{y}_1 = (\delta_0, \ldots, \delta_{N-1})^\top$, the normalized spectrum of $\mathcal{G}_2$ be $\boldsymbol{\delta'} = U^T \Delta \boldsymbol{y}_2 = (\delta'_0, \ldots, \delta'_{N-1})^\top$.

Suppose $1 \le i < N-1$ is odd, $\delta_i = \frac{1}{\sqrt{N/2}} \sum_{i=0}^{N-1} (-1)^i a_{ki}$, $\delta'_i = \frac{1}{\sqrt{N/2}} \sum_{i=0}^{N-1} a_{ki}$ and $\theta = \frac{\pi(i+1)}{N}$, then:

$$
\begin{aligned}
\delta'_i - \delta_i &= \frac{1}{\sqrt{N/2}} \sum_{n=1}^{N/2} \sin\left(\frac{\pi(i+1)(2n-1)}{N}\right) \\
&= \frac{1}{\sqrt{N/2}} \sum_{n=1}^{N/2} \sin\left((2n-1)\theta\right) \\
&= \frac{1}{\sqrt{N/2}} \frac{\sum_{n=1}^{N/2} \sin(\theta) \sin\left((2n-1)\theta\right)}{\sin(\theta)} \\
&= \frac{1}{\sqrt{N/2}} \frac{\sum_{n=1}^{N/2} \left(\cos\left(2n-2\right)\theta\right) - \cos\left(2n\theta\right)}{2\sin(\theta)} \\
&= \frac{1}{\sqrt{N/2}} \frac{\cos(0) - \cos(N\theta)}{2\sin\theta} = 0
\end{aligned}
$$

Therefore, for odd $i$ and $1 \le i \le N-2$, $\delta_i = \delta'_i$. The conclusion can be generalized to even $i$ and $1 \le i \le N-2$ using the same method.

For $i = 0$ and $i = N-1$, we have $\delta_0 - \delta'_0 = -\sqrt{N}$, $\delta_{N-1} - \delta_{N-1} = -\sqrt{N}$. The spectral gap between $\mathcal{G}_1$ and $\mathcal{G}_2$ is:

$$
\begin{aligned}
L_g(\mathcal{G}_1) - L_g(\mathcal{G}_2) &= \sum_{i=0}^{N-1} \frac{2(\delta_i - \delta'_i)g(\lambda_i)}{\sqrt{\sum_{j=0}^{N-1} g(\lambda_j)^2}} \\
&= \frac{2\sqrt{N}g(0) - 2\sqrt{N}g(2)}{\sqrt{\sum_{j=0}^{N-1} g(\lambda_j)^2}}
\end{aligned}
$$

Therefore, the necessary condition for the spectral gap to be 0 is $g(0) = g(2)$. $\qquad \square$

## B  Dataset Details

### B.1  Synthetic Datasets.

We conduct cSBM datasets following [2] in the inductive setting. Denote a cSBM graph $\mathcal{G}$ as $\mathcal{G} \sim \text{cSBM}(n, f, \lambda, \mu)$, where $n$ is the number of nodes, $f$ is the dimension of features, and $\lambda$ and $\mu$ are hyperparameters respectively controlling the proportion of contributions from the graph structure and node features.

We assume the number of classes is 2, and each class is of the same size $n/2$. Each node $v_i$ is assigned with a label $y_i \in \{-1, +1\}$ and an $f$-dimensional Gaussian vector $x_i = \sqrt{\frac{\mu}{n}} y_i u + \frac{Z_i}{\sqrt{f}}$, where $u \sim N(0, I/f)$ and $Z$ is a random noise term.

Assume the generated graph is of average degree $d$, and denote the adjacency matrix as $A$. The graph structure of the cSBM graph is:

$$\mathbb{P}\left[\mathbf{A}_{ij} = 1\right] = \begin{cases} \frac{d + \lambda\sqrt{d}}{n} & \text{if } v_i v_j > 0 \\ \frac{d - \lambda\sqrt{d}}{n} & \text{otherwise.} \end{cases}$$

The parameter $\Phi$ discussed in the experiments is in the form of $\Phi = \arctan(\frac{\lambda}{m\mu}\sqrt{\frac{n}{f}}) * \frac{2}{\pi}$, where $m > 0$ is a constant. A larger $|\Phi|$ reflects a larger $\lambda$ over $\mu$, that is the proportion of information from the graph structure is larger.

In practice, we choose $n = 3000, f = 2000, d = 5, m = \frac{3\sqrt{3}}{2}$ for all graphs. The choices of $\lambda$ and $\mu$ and the resulting homophily ratio are listed in Table 1. As discussed in [3], only the hyperparameters $\mu$ and $\lambda$ that satisfy $\lambda^2 + \frac{\mu^2 f^2}{n^2} > 1$ are guaranteed to generate informative cSBM graphs. As presented in Table 1, all our settings satisfy the need.

Table 1: Statistics of cSBM datasets.

| $\Phi$ | +0.75 | +0.50 | -0.50 | -0.75 |
|---|---|---|---|---|
| $\lambda$ | 1.90 | 1.46 | -1.46 | -1.90 |
| $\mu$ | 0.37 | 0.69 | 0.69 | 0.37 |
| Homophily Level | 0.92 | 0.82 | 0.18 | 0.08 |

## C  Experiment Details

### C.1  Experimental Device

Experiments are conducted on a device with an NVIDIA TITAN V GPU (12GB memory), Intel(R) Xeon(R) Silver 4114 CPU (2.20GHz), and 1TB of RAM.

### C.2  Model Architectures

For GPRGNN, GCNII, GNNGuard and ProGNN, we rely on the officially released code. For FAGCN, we implement the method with Pytorch Geometric(PyG) based on the released code. For H2GCN, we rely on the PyG version implemented by [12]. Defense models are based on the DeepRobust Library implemented versions[11]. Other methods are based on the PyG implemented versions [4]. The URL and commit number are presented in Table 2).

Table 2: Code & commit numbers.

| | URL | Commit |
|---|---|---|
| GPRGNN | https://github.com/jianhao2016/GPRGNN | eb4e930 |
| ProGNN | https://github.com/ChandlerBang/Pro-GNN | c2d970b |
| GNNGuard | https://github.com/mims-harvard/GNNGuard | 88ab8ff |
| GCNII | https://github.com/chennnM/GCNII | ca91f56 |
| FAGCN | https://github.com/bdy9527/FAGCN | 23bb10f |
| H2GCN | https://github.com/CUAI/Non-Homophily-Large-Scale | 281a1d0 |

### C.3  Hyperparameter settings

**Node classification on cSBM Datasets & Common datasets.**    For all models, we use early stopping 200 with a maximum of 1000 epochs. All hidden size of layers is set to 64. We use the Adam optimizer and search the optimal leaning rate over {0.001, 0.005, 0.01, 0.05} and weight decay {0.0,

0.0005}. For all models, the linear dropout is searched over {0.1, 0.3, 0.5, 0.7, 0.9}. For the model-specific hyperparameters, we refer to the optimal hyperparameters reported in corresponding papers. For MLP, we include 2 linear layers. For GCN and H2GCN, we set the number of convolutional layers to be 2. For GAT, we use 8 attention heads with 8 hidden units each in the first convolutional layer, and 1 attention head and 64 hidden units in the second convolutional layer. For FAGCN, we search the number of layers over {2, 4, 8}, $\epsilon$ over {0.3, 0.4, 0.5}. For GCNII, we set $\lambda = 0.5$ and search the number of layers over {8, 16, 32}, $\alpha$ over {0.1 0.3 0.5}. For GPRGNN and EvenNet, we set the number of linear layers to be 2, the learning rate for the propagation layer to be 0.01, and $\alpha = 0.1$ for initialization. For both models, we search the dropout rate for the propagation layer over {0.1, 0.3, 0.5, 0.7} and the order of graph filter over {4, 6, 8, 10}.

**Against adversarial attacks.** For the poisson attacks, we use a 2-layer GCN as the surrogate model. We use the strongest variant of Metattack, which is "Meta-Self" as the attack strategy. For the defense models, we carefully follow their provided guidelines of hyperparameter settings and use the optimal hyperparameters as they reported. For GNNGuard, we use the official implementation GCNGuard with a threshold of 0.1. For other models, we use the Adam optimizer with a learning rate of 0.01, weight decay of 0.0005, and a dropout rate of 0.5. For FAGCN, we use 8 convolutional layers and a fixed $\epsilon = 0.3$. For GCNII, we use 16 convolutional layers and a fixed $\alpha = 0.2$. For GPRGNN and EvenNet, we set the order of graph filter $K$ to be 4 in the DICE attack. In the poison attacks, the PPR initialization $\alpha$ for GPRGNN and EvenNet is searched over {0.1, 0.2, 0.5, 0.9} and $K$ is set to be 10.

For all models, we use early stopping 30 with a maximum of 200 epochs. Other hyperparameters are kept the same as the ones in the node-classification experiments.

## C.4 Additional Defense Results

**Homophily gap** We include the homophily gap between training and test graph for Citeseer and ACM datasets in Fig 1 and 2. The homophily gaps of all attacks on all datasets grow larger as the perturb ratio increases.

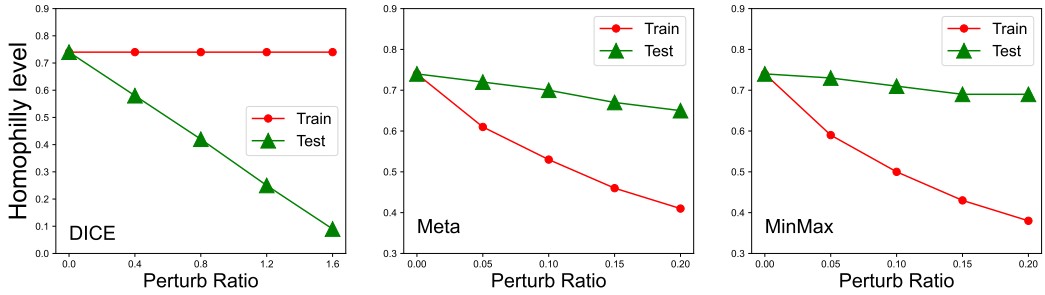

Figure 1: Homophily level of training graphs and test graphs on Citeseer after attacks.

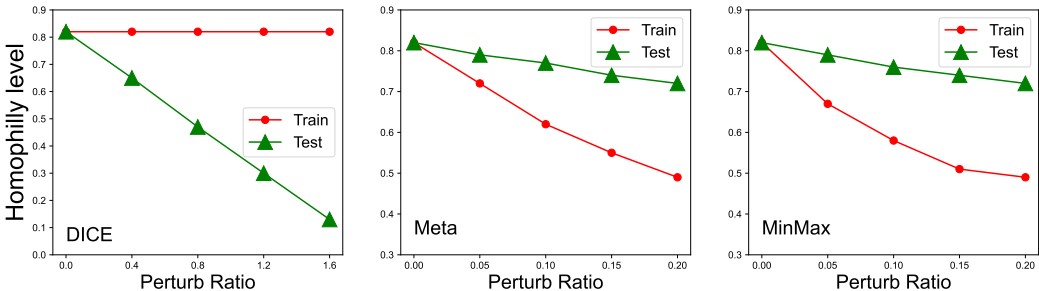

Figure 2: Homophily level of training graphs and test graphs on ACM after attacks.

**Additional Experiments about Defense against Poison Attacks.** Similar to DICE attacks, we provide the performance of GNN models under poison attacks of different perturb ratios. The results

are presented in Figure 3 and 4. In most cases, EvenNet achieves SOTA with fewer introduced parameters.

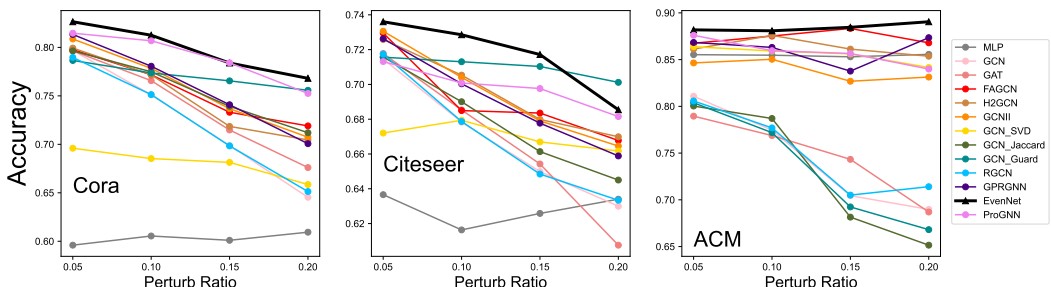

Figure 3: Meta attack on three homophilic datasets. EvenNet is marked with "△".

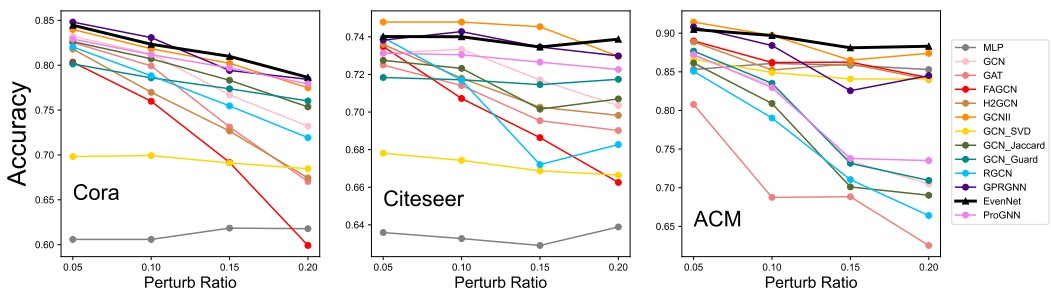

Figure 4: MinMax attack on three homophilic datasets. EvenNet is marked with "△".

# D  Spectral Methods under Perturbations

## D.1  More Graph filters

Spectral methods have gained plenty of attention these years. Besides GPRGNN, we additionally include BernNet [7] and pGNN [5] which are advanced spectral GNNs for comparison. We run the experiments of spectral GNNs against Meta, MinMax attacks, and the evasion DICE on dataset ACM. We tune the hyperparameters of these methods using the same search space in corresponding papers. The results are summarized below:

Table 3: Defense performance under Metattack with perturb ratio 20%.

| Methods \Dataset | Cora | Citeseer | ACM |
|---|---|---|---|
| GPRGNN | $76.27 \pm 1.43$ | $69.63 \pm 1.53$ | $88.79 \pm 2.21$ |
| pGNN | $72.68 \pm 2.38$ | $67.20 \pm 1.30$ | $\mathbf{89.92 \pm 0.66}$ |
| BernNet | $74.38 \pm 2.00$ | $67.93 \pm 1.33$ | $87.82 \pm 0.98$ |
| EvenNet | $\mathbf{77.74 \pm 0.82}$ | $\mathbf{71.03 \pm 0.97}$ | $\underline{89.78 \pm 0.90}$ |

Table 4: Defense performance under MinMax attack with perturb ratio 20%.

| Methods \Dataset | Cora | Citeseer | ACM |
|---|---|---|---|
| GPRGNN | $77.18 \pm 1.37$ | $72.81 \pm 0.78$ | $88.24 \pm 1.28$ |
| pGNN | $77.06 \pm 1.32$ | $72.22 \pm 0.53$ | $88.96 \pm 0.56$ |
| BernNet | $69.10 \pm 1.07$ | $67.82 \pm 0.79$ | $87.79 \pm 0.41$ |
| EvenNet | $\mathbf{78.40 \pm 1.26}$ | $\mathbf{73.51 \pm 0.60}$ | $\mathbf{89.80 \pm 0.46}$ |

Compared with spatial methods, spectral methods which handle both homophily and heterophily are generally more robust. Nevertheless, EvenNet still holds superiority against other spectral methods

Table 5: Defense performance under DICE attack on ACM dataset with different perturb ratios.

| Methods \Perturb Ratios | 0.4 | 0.8 | 1.2 |
|---|---|---|---|
| GPRGNN | $79.31 \pm 1.05$ | $73.21 \pm 1.93$ | $63.41 \pm 9.21$ |
| pGNN | $86.67 \pm 0.91$ | $84.55 \pm 2.66$ | $81.62 \pm 2.32$ |
| BernNet | $86.37 \pm 3.30$ | $82.79 \pm 3.74$ | $81.90 \pm 4.46$ |
| EvenNet | $\mathbf{89.24 \pm 0.52}$ | $\mathbf{88.26 \pm 0.82}$ | $\mathbf{88.67 \pm 0.64}$ |

when faced with large homophily changes. In the evasion DICE attack, where the homophily gap is directly injected between training and test graphs, the superiority of EvenNet is apparent.

## D.2 Graph Filters under Random Attacks

We analyze the performance of graph filters under homophily change in the main body of the paper and show that graph filters suffer from performance degradation if there is a large homophily gap between training and test graphs. We now discuss cases where a homophily gap is not huge after the graph structure is perturbed, for example, when the graph is under random attacks for both training and test sets.

We conduct Random Attacks on datasets Cora and Citeseer for the spectral methods. In Random attacks, we randomly delete and add edges from/to the graph (we choose to delete or add with equal probability), and train graph filters on the perturbed graph. A large homophily gap does not exist since the deleted/added edges are randomly chosen on the whole graph.

We also include a spatial method EGCNGuard from [1], which is an efficient version of GNNGuard for comparison. The results are summarized in Table 6 and Table 7:

Table 6: Defense performance under Random attacks on Cora

| Methods \Perturb Ratio | 20% | 40% | 60% |
|---|---|---|---|
| GPRGNN | $82.62 \pm 0.31$ | $78.86 \pm 0.60$ | $76.68 \pm 0.12$ |
| pGNN | $\mathbf{83.54 \pm 0.19}$ | $\mathbf{80.52 \pm 0.52}$ | $\mathbf{77.18 \pm 0.49}$ |
| BernNet | $78.40 \pm 3.11$ | $73.69 \pm 3.20$ | $69.34 \pm 1.48$ |
| EvenNet | $82.37 \pm 0.49$ | $78.95 \pm 0.47$ | $76.01 \pm 0.71$ |
| EGCNGuard | $77.62 \pm 1.40$ | $75.77 \pm 1.46$ | $73.20 \pm 1.01$ |

Table 7: Defense performance under Random attacks on Citeseer

| Methods \Perturb Ratio | 20% | 40% | 60% |
|---|---|---|---|
| GPRGNN | $\mathbf{73.45 \pm 0.81}$ | $70.25 \pm 0.46$ | $69.72 \pm 0.79$ |
| pGNN | $72.61 \pm 0.93$ | $\mathbf{72.52 \pm 0.66}$ | $\mathbf{70.36 \pm 1.44}$ |
| BernNet | $66.98 \pm 1.25$ | $66.47 \pm 0.73$ | $66.80 \pm 0.39$ |
| EvenNet | $73.01 \pm 0.68$ | $71.30 \pm 0.78$ | $69.66 \pm 0.50$ |
| EGCNGuard | $72.12 \pm 0.82$ | $69.61 \pm 1.42$ | $66.98 \pm 2.60$ |

Although EvenNet is designed based on homophily generalization, we could all spectral methods including EvenNet are quite robust under random attacks, reflecting good stability under random perturbations of graph structures. For a more comprehensive analysis of the performance of graph filters under random perturbations, we refer the readers to stability theory, where the bound of change in output of graph filters is discussed [6, 9, 10]. EvenNet as a spectral method holds the stability property as well.

## E   Scability to large graphs

In this section, we try to run EvenNet on a larger dataset to verify its efficiency. For comparison, we include vanilla GCN and the efficient implementation of GNNGuard EGCNGuard from [1]. Notice that we could not run GNNGuard and ProGNN for their $O(N^2)$ space complexity.

In Table E and Table E, we present the performance and running time of EvenNet on dataset ogbn-arxiv against Random attacks. We could see that EvenNet uses almost the same time as a 2-layer GCN with the same hidden size. And EGCNGuard of $O(|E|)$ space complexity is still 3x slower than EvenNet in practice.

Table 8: Defense performance under Random attacks on dataset ogbn-arxiv.

| Methods \Perturb Ratio | 20% | 40% | 60% |
|---|---|---|---|
| GCN | 64.07 | 60.96 | 58.45 |
| EGCNGuard | **64.52** | 60.81 | 57.18 |
| EvenNet | 64.18 | **61.20** | **58.97** |

Table 9: Computational time under Random attacks on dataset ogbn-arxiv.

| Methods | Avg. training time per epoch (s$\times 10^{-3}$) |
|---|---|
| GCN | 0.253 |
| EGCNGuard | 1.181 |
| EvenNet | 0.38 |

## F    Defense against Graph Injection Attacks

In the main body of the paper, we mainly focus on graph modification attacks, which are graph structural attacks that add/remove edges to/from the existing graph. Another line of graph structural attack is graph injection attacks (GIAs), where new nodes are injected into the graph with generated features and form connections with existing nodes on the graph. According to [1], GIAs significantly degrade the performance of GNNs by injecting only a few nodes with limited budgets. The authors also state that injecting nodes with suspicious features that result in homophily inconsistency helps in enhancing the ability to attack.

Following [1], we apply non-targeted GIAs with Harmonious Adversarial Objective(HAO) including PGD+HAO, AGIA+HAO, TDGIA+HAO as the attack methods. The authors claim that GIAs with HAO cause larger homophily gaps between training and test graphs. We test the defense performance of EvenNet on dataset grb-cora, grb-citeseer [13] and Arxiv [8]. We use the same budgets as in [1], which is reported in Table F. We compare EvenNet including Layernorm in MLP layers with different combinations of EGCNGuard with Layernorm and LNi operation. (Layernorm is shown to be effective against GIAs.) We set the threshold for EGCNGuard as 0.1. We set the order of EvenNet $K$ as 2 and tune the PPR-like initialization of EvenNet $\alpha$ over $\{0.1, 0.2, 0.5, 0.9\}$

The results are summarized below in Table F and Table F. We are not able to run AGIA+HAO on Arxiv dataset due to resource limitations.

Table 10: Fixed budges of GIA attacks.

| Datasets \Perturb Ratio | Inject Nodes | Degree |
|---|---|---|
| Cora | 60 | 20 |
| Citeseer | 90 | 10 |
| Arxiv | 1500 | 100 |

Table 11: Defense performance of EvenNet under GIA with fixed budgets.

| Methods \Datasets | grb-Cora | grb-Citeseer | arxiv |
|---|---|---|---|
| PGD+HAO | 76.24 | 71.68 | 59.04 |
| AGIA+HAO | 75.25 | 71.26 | – |
| TDGIA+HAO | 77.23 | 70.85 | 55.12 |

Notice that although GIAs include structural modifications by adding edges between injected nodes and existing nodes, GIAs are not pure structural attacks. The injected node features are usually

Table 12: Defense performance of EGCNGuard under GIA with fixed budgets.

| Methods \Datasets | grb-Cora | grb-Citeseer | Arxiv |
|---|---|---|---|
| PGD+HAO | 75.50 | 58.10 | 69.37 |
| AGIA+HAO | 72.88 | 56.32 | – |
| TDGIA+HAO | 73.75 | 58.10 | 51.23 |

learnable, and therefore suspicious node features together with structural perturbations are included, which is beyond the scope of the paper. We choose to leave designing robust spectral methods under feature perturbations to future works. Notwithstanding GIAs are somehow out of the scope of EvenNet, we can see that EvenNet is still competitive against strong spatial baselines in Table F and Table F, which verifies the ability of EvenNet under homophily change.

## G  Defense on Heterophilic Graphs

In the main body of the paper, we conduct attacks mainly on homophilic graphs. In this section, we add experiments about the MinMax attack on heterophilic datasets chameleon and squirrel with GCN being the surrogate model. We set the perturb ratio to be 20%. For the hyperparameters, we search the learning rate over {0.01, 0.05} and set weight decay to be 0 for all models. We set the threshold in EGCNGuard to 0.1. The number of layers used in H2GCN and FAGCN is set to be 2, and $\epsilon$ is searched over {0.3, 0.4, 0.5} for FAGCN. For GPRGNN and EvenNet, we set $\alpha = 0.1$ for PPR initialization and $K = 10$. The results are summarized below:

Table 13: The average test accuracy against MinMax attack on heterophilic graphs over 5 different splits.

| Method & Dataset | Chameleon | Squirrel |
|---|---|---|
| MLP | $48.84 \pm 1.66$ | $30.31 \pm 1.25$ |
| GCN | $49.93 \pm 0.70$ | $31.16 \pm 2.19$ |
| EGCNGuard | $45.34 \pm 2.80$ | $27.34 \pm 0.90$ |
| H2GCN | $51.42 \pm 1.31$ | $28.41 \pm 1.08$ |
| FAGCN | $49.98 \pm 1.27$ | $\mathbf{33.64 \pm 1.10}$ |
| GPRGNN | $50.42 \pm 0.83$ | $32.47 \pm 1.36$ |
| EvenNet | $\mathbf{52.87 \pm 1.88}$ | $33.21 \pm 0.96$ |

We can see from the results that EvenNet is still effective in defense against attacks on heterophilic graphs. Yet, we did not focus on heterophilic datasets as GNNs already perform badly on them, which is also a reason why current attacks mainly focus on homophilic graphs. On the Squirrel dataset, the performance of GNNs is only slightly higher than MLP, reflecting an almost useless graph structure.