# OpenReview forum: "EvenNet: Ignoring Odd-Hop Neighbors Improves Robustness of Graph Neural Networks"
_NeurIPS.cc/2022/Conference — NeurIPS 2022 Accept_

### Official Review · Reviewer_FdNz · 2022-07-07

**Rating:** 6
**Confidence:** 4
**Soundness:** 3 good
**Presentation:** 3 good
**Contribution:** 3 good

**Summary:**

This paper proposes to ignore odd-hop neighbors to improve the robustness of graph neural network. The authors analyze the influence of odd and even-hop neighbors to the graph filters and reveal that using only even-hop neighbors can improve the robustness of spectral graph neural networks. Based on the theoretical findings, the authors propose EvenNet and conduct several experiments to validate the effectiveness of EvenNet.

**Questions:**

After the rebuttal, the authors resolve most of my concerns. Thus I raise my ratings to 6.

===========
The motivation of the paper is confusing:
- The EvenNet design seems to be heuristic, which tends to assume the task is binary thus the neighbors of heterophilic neighbors tend to have homophilic to the central nodes. I am not sure why this heuristic is a more general topological information beyond homophily.

- Furthermore, the theoretical analysis throughout the paper including the synthetic experiments inherent the strong assumption on the problem structure, i.e., binary classification. Under binary classification, incorporating even-hop neighbors is straightforward as shown in Figure 1 by the author, which may weakens both the significance and novelty of the paper.

There are several missing important comparisons and experiments:
- Comparisons between existing spectral GNNs which are also designed to work on both homophily and heterophily GNNs are missing, e.g., Fu et al., 2021[1].
- The paper focuses on a inductive setting and discusses the homophily changes under adversarial attacks. In fact, Chen et al., 2022[5] established the same discussion in this setting under a more ``powerful’’ graph adversarial attack, i.e., Graph Injection Attack (GIA) [2,3,4,5]. More discussions and comparisons are required to distinguish the contributions of the paper, and more importantly, the effectiveness of EvenNet (See more details in the next point).
- In fact, most of existing graph adversarial attacks are limited to small scale graphs [6] in contrast to GIA which works on both small and large scale graphs [4,5], making the evaluations less convincing [4,5,6]. It would strengthen the claims and further improve the impact of the paper if the authors could validate the high robustness of EvenNet in large scale graphs.
- The authors claim the proposed method enjoys cheaper computational overhead than existing robust GNNs, yet did not provide any justifications.

The writing of the paper also misses several important details. Here are few:
- In Theorem 1, how the expectation is taken given the subscript as H and inputs \PI?
- In Line 150, why the regression problem of \alpha and \beta can be seen as linear?
- In Theorem 3, what is the relationship between homophily/heterophily and the non-increasing/non-decreasing \alpha?
- In experiments, how the threshold in GNNGuard is chosen?
- In all tables of the experiments, why are the standard variances are missing?
- How robust are EvenNet under Metattack with different budgets?

References:

[1] Guoji Fu, Peilin Zhao, Yatao Bian. p-Laplacian Based Graph Neural Networks. arXiv 2021.

[2] Wei Jin, Yaxin Li, Han Xu, Yiqi Wang, Shuiwang Ji, Charu Aggarwal, Jiliang Tang. Adversarial Attacks and Defenses on Graphs: A Review, A Tool and Empirical Studies. SIGKDD Explorations 2020.

[3] Lichao Sun, Yingtong Dou, Carl Yang, Ji Wang, Philip S. Yu. Adversarial Attack and Defense on Graph Data: A Survey. arXiv 2020.

[4] Qinkai Zheng, Xu Zou, Yuxiao Dong, Yukuo Cen, Da Yin, Jiarong Xu, Yang Yang, Jie Tang. Graph Robustness Benchmark: Benchmarking the Adversarial Robustness of Graph Machine Learning. NeurIPS 2021 Datasets and Benchmark Track.

[5] Yongqiang Chen, Han Yang, Yonggang Zhang, Kaili Ma, Tongliang Liu, Bo Han, James Cheng. Understanding and Improving Graph Injection Attack by Promoting Unnoticeability. ICLR 2022.

[6] Simon Geisler, Tobias Schmidt, Hakan Şirin, Daniel Zügner, Aleksandar Bojchevski, Stephan Günnemann. Robustness of Graph Neural Networks at Scale. NeurIPS 2021.



**Limitations:**

Could not find the corresponding discussion in the paper.

**Strengths And Weaknesses:**

*Originality & Significance*: Improving the robustness of graph neural networks is of great importance to the community. The proposed solution enjoys lots of theoretical insights from the spectral graph neural networks and the balance theory. However, the motivation is confusing and the empirical evidences may not be sufficient to support the authors’ claims.

*Quality*: Despite of certain theoretical supports for the design of EvenNet. The theoretical analysis tend to put strong assumptions on the underlying graph homophily/heterophily. The empirical analysis might not be sufficient, as several important experiments are missing. See the questions for more details.

*Clarity*: Overall, the paper is well-organized, however, there are several important details are missing, making readers hard to follow.

---

> ### Author Response · Authors · 2022-08-02
> **Our Response (I)**
>
> Thanks for your comprehensive and insightful feedback!
>
> **Q1:** The motivation seems heuristic as it arises from binary tasks.
>
> **A1:** We present the binary friend-enemy network in the introduction as a toy example to show that even-hop neighbors are somehow more robust than odd-hop neighbors. We provide a discussion of multi-class cases in **Appendix A.1** from the spatial perspective.
> The primary motivation for this paper emerges from the spectral domain, where EvenNet is based on a polynomial basis symmetric about $\lambda=1$, yielding a band-reject filter.
> However, multi-class heterophily in the spectral domain is a challenging subject and has received very little research. Recent works that focus on spectral methods, such as [1] and [2], start with binary tasks or 2-coloring cases in the analysis.
> At the same time, as pointed out in [1], an $m$-classes task can be regarded as $m$ binary-class tasks, which means the conclusions that arise from binary tasks are still inspiring.
> Finally, the empirical results on multi-class datasets also verify the effectiveness of EvenNet.
> So we decide to leave a more general analysis of the multi-class problem in the spectral domain to future works.
>
> **Q2:** Comparison against advanced spectral methods is missing.
>
> **A2:** Thanks for your advice about adding spectral benchmarks.
> We have added experiments about pGNN[3] and BernNet[4] in **Appendix D.1**. (please refer to the revised version of the Appendix)
> Here, we report the results of spectral methods under Metattack with 20\% edges the budges.
> (The results of GPRGNN are higher than we reported in the previous version, we will update them in the revised version.)
> EvenNet is still effective compared with advanced spectral methods.
> A more general discussion of spectral methods under perturbations is also provided in **Appendix D.2**.
>
> **Defense performance under Metattack with perturbing ratio 20\%**
> | Methods \ Dataset | Cora | Citeseer | ACM |
> | :-----           | :----: | :----: |:----: |
> | GPRGNN | 76.27 $\pm$ 1.43 | 69.63 $\pm$ 1.53 | 88.79 $\pm$ 2.21 |
> | pGNN | 72.68 $\pm$ 2.38 | 67.20 $\pm$ 1.30 | **89.92 $\pm$ 0.66** |
> | BernNet | 74.38 $\pm$ 2.00 | 67.93 $\pm$ 1.33 |  87.82 $\pm$ 0.98 |
> | EvenNet | **77.74 $\pm$ 0.82** | **71.03 $\pm$ 0.97** | 89.78 $\pm$ 0.90 |
>
> **Q3:** Comparison against graph injection attacks (GIAs) are missing.
>
> **A3:** We appreciate the advice about considering more attack methods.
> In this work, we mainly focus on graph structure attacks, as we discussed in the introduction section. GIAs inject nodes with suspicious features and are thus beyond the scope of our work, so we didn't include GIA in the previous version.
> That said, we have added the experiments of EvenNet under GIA attacks to the revised Appendix.
> Here we present the results of EvenNet with layernorm under GIAs in the below table.
> More details are included in the **Appendix F**.
>
> **EvenNet+layernorm against GIAs**
> |Methods \ Datasets | grb-Cora | grb-Citeseer | arxiv |
> | :----- | :----: | :----: |:----: |
> |PGD+HAO | 75.50 | 58.10  |  69.37|
> |AGIA+HAO | 72.88 | 56.32  |  \ |
> |TDGIA+HAO | 73.75 | 58.10  |  51.23|
>
> Even though EvenNet is not designed for GIAs, it can be integrated with other defense methods and is still competitive under GIAs.
>
> **Q4:** The results on large datasets are missing.
>
> **A4:** Theoretically, EvenNet takes O($E$) space complexity ( $E$ is the number of edges) since it only needs to store the adjacency matrix. Compared with the original GNNGuard and proGNN that achieve O($N^2$) space complexity, EvenNet can easily be scaled to larger datasets, e.g. arxiv.
> Here we present the results and the average training time per epoch of GCN, EvenNet, and EGCNGuard(An efficient GNNGuard from [5]) under Random attack on arxiv.
> We are not able to run Metattack \| MinMax attacks on arxiv since they are not scalable.
>
> | Methods \ Perturb Ratio | 20\% | 40\% | 60\%|
> | :----- | :----: | :----: |:----: |
> |   GCN | 64.07  |  60.96  | 58.45 |
> |EGCNGuard | **64.52** | 60.81  | 57.18 |
> |EvenNet | 64.18 | **61.20**  | **58.97** |
>
> | Methods | Avg. training time per epoch (s$\times 10^{-3}$) |
> | :----- | :----: |
> | GCN | 0.253 |
> | EGCNGuard | 1.181 |
> | EvenNet | 0.38 |
>
> We can see that EvenNet is scalable to large graphs with low time consumption.
> We have added more detailed comparisons and experiments in **Appendix E** that demonstrate the efficiency of EvenNet.

---

> ### Author Response · Authors · 2022-08-02
> **Our Response (II)**
>
> **Q5:** How the expectation is taken given the subscript as $\mathcal{H}$ and inputs $\Pi$?
>
> **A5:** Suppose a graph contains a fixed number of nodes, but node labels and edges are varied, which results in a series of graphs with different homophily levels.  We assume the 1-homophily levels $\mathcal{H}$ of these graphs follow a uniform distribution in [-1,1], the expectation is taken  on these graphs given filter $g$. The detailed proof is included in **Appendix A.1**
>
> **Q6**: The linear correlation between $\alpha$ and $\beta$? What is the relationship between homophily/heterophily and the non-increasing/non-decreasing $\alpha$?
>
> **A6:** The linear assumption of $\alpha$ and $\beta$ in line 150 were given as an example, corresponding to the linear correlations between labels and input signals. We believe the discussion of the simple linear case would offer a good intuition of how the SRL works for graph filters.
>
> In fact, Theorem 3 still holds when $\alpha_i^2$ is larger than $\alpha_{n-i}^2$, which is a special case of the monotonicity requirement. On certain graphs such as ring graphs and grid graphs, homophily means more frequency components on lower frequencies and vice versa. (This can be verified by explicitly computing their eigenvectors of Laplacian, Theorem 2 offers intuition about this fact.)
> We admit that the non-increasing/non-decreasing $\alpha$ is a relatively strong assumption even on binary tasks. A more general theorem relies on a deeper understanding of the graph signal distribution from the spectral domain, which is currently lacking and beyond the scope of this paper. We will clarify this limitation of our theorems in the revised version of the paper.
>
> **Q7:** How the threshold in GNNGuard is chosen? / In all tables of the experiments, why are the standard variances are missing? How robust is EvenNet under Metattack with different budgets?
>
> **A7:** For the hyperparameters of GNNGuard, we refer to the implementation of the official code and use threshold 0.1 for all experiments.
> We didn't include standard variances due to space limitations.
> We report some of the standard variances of EvenNet in **Appendix D**.  The results of EvenNet under Metattack with different budgets are given in **Appendix C.4**.
>
> If you still have any other questions about this work, we are looking forward to your reply!
>
> [1] Chen Zhixian, Tengfei Ma, and Yang Wang. "When Does A Spectral Graph Neural Network Fail in Node Classification?." arXiv preprint, 2022.
>
> [2] Yimeng Min, Frederik Wenkel, Guy Wolf. "Scattering GCN: Overcoming Oversmoothness in Graph Convolutional Networks." NeurIPS 2020
>
> [3] Fu Guoji, Peilin Zhao, and Yatao Bian. "$p$-Laplacian Based Graph Neural Networks." ICML 2022.
>
> [4] He Mingguo, Zhewei Wei, and Hongteng Xu. "BernNet: Learning arbitrary graph spectral filters via Bernstein approximation." NIPS 2021.
>
> [5] Yongqiang Chen, Han Yang, Yonggang Zhang, Kaili Ma, Tongliang Liu, Bo Han, James Cheng. Understanding and Improving Graph Injection Attack by Promoting Unnoticeability. ICLR 2022.

---

> > ### Comment · Reviewer_FdNz · 2022-08-04
> > **Reply to authors**
> >
> > I thank the authors for the detailed reply. They resolve most of my concerns. I'll reconsider the ratings.

---

> > > ### Author Response · Authors · 2022-08-04
> > > **Reply to Reviewer FdNz**
> > >
> > > Thank you for your time and effort!

---

### Official Review · Reviewer_Pbwq · 2022-07-10

**Rating:** 7
**Confidence:** 4
**Soundness:** 2 fair
**Presentation:** 3 good
**Contribution:** 3 good

**Summary:**

This work designs a simple and robust GNN model, EvenNet, in the spectral domain. EvenNet discards messages from odd-order neighbors. Authors analyze this method theoretically. In the spatial domain, authors define homophily degree, which reflects the average possibility of deriving a node's label from its k-hop neighbors, and then prove that even-order filter can benefit from a lower variance of homophily degree. In the spectral domain, the author proves that an even-order filter can achieve a lower loss gap between the homophilic training graph and the heterophilic test graph. EvenNet exhibits generalization ability across homophily and robustness to many attacks in experiments. Moreover, it also keeps competitive performance and low time complexity.

**Questions:**

Why not use $\phi_{test}\neq\pm\phi_{train}$ in experiments? For example, can we use a setting like $\phi_{test}=0.3, \phi_{train}=0.75$?


**Limitations:**

Not related.

**Strengths And Weaknesses:**

Strength:

1. The method is simple but novel.
2. Experiments are persuasive.
3. The paper is well-written.

Weakness

Theoretical analysis is largely based on heuristics. In the spatial domain, whether the homophily degree change represents the effects of homophily change on GNN is doubtful. In the spectral domain, the analysis is based on too many unnatural assumptions like 1-dimensional feature, non-increasing and non-decreasing $\alpha$, and ring.

---

> ### Author Response · Authors · 2022-08-02
> **Our Response**
>
> Thanks for your appreciation of our work!
>
> **Q1:** Why not use different $\phi_{train}$ $\ne$ $\pm\phi_{test}$ in experiments?
>
> **A1:** We carefully follow the settings for cSBM models in [1].
> From the discussion of the synthetic cSBM models in [1], we could see that $|\phi|$ controls the proportion of helpful information between node features and the graph structure.
> A larger $|\phi|$ reflects a larger proportion of information from the graph structure.
>
> To ensure fairness in the generalization across graphs of different homophily, we need to maintain the proportion of information that comes from the graph structure the same.
> The only difference should be how the information is revealed on the training and test graphs - whether they are of the same homophily or opposite homophily.
> If we train a model with helpful node features and generalize it to a graph with suspicious node features, it becomes hard to tell whether the performance of the model is being affected by the change in the graph structure or the change of node features.
> So in the paper, we decide to use $\phi_{train}$ $=$ $\pm\phi_{test}$.
> We also include a more detailed discussion of the cSBM model in Appendix B.2.
>
> We have also updated a revised version of the Appendix.
> In Appendix D, we compare EvenNet with more advanced spectral methods and run experiments of EvenNet under the Random Attack where a homophily gap does not exist.
> Our proposed method is still shown effective, and we provide a discussion in the field of filter stability as an explanation.
> In Appendix E, we theoretically analyze the scalability of EvenNet and run EvenNet on a large dataset ogb-arxiv under Random Attack.
> We report the average training time per epoch that verifies the efficiency of EvenNet.
> In Appendix F, we provide the results of EvenNet under Graph Injection Attacks(GIAs).
> We can see that even though EvenNet is not designed for GIAs, it can be integrated with other defense methods and is still competitive.
>
> If you have any other questions, we are looking forward to your response!
>
> [1] Chien Eli, et al. "Adaptive universal generalized PageRank graph neural network." ICLR, 2021

---

### Official Review · Reviewer_cSNX · 2022-07-11

**Rating:** 5
**Confidence:** 4
**Soundness:** 3 good
**Presentation:** 3 good
**Contribution:** 2 fair

**Summary:**

This paper studies GNN robustness against homophily changes (e.g., whether direct neighborhoods belong to the same class). It proposes a spectral graph filter that only considers even-polynomial signals, so that it’s robust to odd-hop changes.

**Questions:**

See above

**Strengths And Weaknesses:**

Strengths

The paper gives clear motivation of why only considering odd-order might tackle homophily changes.
The paper gives theoretical analysis from both spatial and spectral perspectives.

Weaknesses (or questions)

1. Is EvenNet only effective in homophily change?

I think the paper is based on an assumption that most perturbation comes from homophily changes, more specifically, label changes of the direct neighborhood. Based on this assumption, taking even-hop and ignoring odd-hop could increase robustness and loss accuracy. However, this is a very specific type of attack (and from my understanding, people mainly study this to show the weakness of vanilla GCN). So I have the following two questions: is Evennet only effective for homophily changes? What if you also consider some general graph attack, such as random perturbation, a trained attacker? Or more specifically, what if we change the association of label correlation at even-hop (say 2) instead of the direct hop? It would be better if the authors could give an analysis to let people know at which scope the proposed method is effective;

2. Will ignoring odd-hop affect performance?

From my understanding of most graph datasets, first-hop neighborhoods contain the most information. If simply ignoring all odd-hop neighbors, will the performance be influenced a lot? Table 3 shows that EvenNet achieves competitive results with many old baselines, but it would be more convincing if the authors could conduct some ablation study, because the proposed approach and baselines could be run in different evaluation settings.

3. Theoretical analysis might not support the claim

Theorem 1 talks about whether an even-order graph filter has lower variation than full-order given different edge homophily. This could not support the author’s claim of “without losing average performance”. If we don’t do graph filtering and only use each node’s own feature the variance is 0, but it doesn't mean this baseline is better than a graph filter.

The result of Theorem 3 is very interesting, however, I could not really get how you estimate L(G), as it should depend on the filter parameter g, which I couldn’t figure out how you get even after reading the appendix. If you have some assumptions about this Theorem, it would be better to clearly state it so people might understand when the conclusion holds.


4. Experiment-wise, it would be better to test on larger-scale and more standard datasets such as OGB. Also, it would be good to show results with real-world distribution shifts instead of only synthetic shifts.

---

> ### Author Response · Authors · 2022-08-02
> **Our Response (I)**
>
> Thanks for your comprehensive and insightful feedback!
>
> **Q1:** Is EvenNet only effective in homophily change?
>
> **A1:** EvenNet is effective under general attacks such as Random Attack as well.
> First of all, we provide a discussion of the robustness of spectral methods under general structural perturbations as the filter stability in **Appendix D.2**. (Please refer to the revised version.)
> EvenNet is stable as a spectral method.
> Secondly, we run spectral methods under the Random attack in **Appendix D.2**, where the results also support the above claims.
>
> Regardless of the stability of graph filters, the lack of constraints on the learned filters could lead to failure in generalizing when encountering homophily change, which is common as a result of advanced graph attack methods.
> And EvenNet is shown helpful to ease the problem.
>
> **Q2:** What if we change the association of label correlation at even-hop (say 2) instead of the direct hop?
>
> **A2:** It is a really good question!
> For a single node, it is possible to inject two-hop heterophily directly.
> However, compare to two-hop homophily, one-hop homophily is more fragile globally.
> In **Appendix A.1**, we discussed the advantages of even-hop neighbors in the presence of structural perturbations, and here we present a straightforward toy example.
>
> Consider the graph A-B-C, where A, B, and C represent nodes and '-' represents an edge. Suppose that A, B, and C each have unique labels.
> By adding edge B-C, the two-hop heterophily of node A is increased.
> However, such an operation does not increase global two-hop heterophily because it decreases node B's two-hop heterophily, where B-C-B becomes a two-hop heterophilic path.
> In contrast, the addition of the edge B-C immediately increases the global one-hop homophily.
> Consequently, it is considerably more difficult to create a two-hop homophily gap on the graphs.
> The table below shows the change in two-hop homophily for learnable attack strategies.
> As can be seen, the two-hop homophily gap is typically smaller than the one-hop homophily gap.
>
> **Homophily gap after Meta/MinMax attacks with 20\% the perturb ratio.**
> | Homophily \ dataset | Meta-Cora | Meta-Citeseer | Meta-ACM | MinMax-Cora | MinMax-Citeseer |MinMax-ACM |
> | :----- | :----: | :----: |:----: |:----: |:----: |:----: |
> |1-hop Train | 0.42 | 0.4| 0.49 | 0.36 | 0.38 | 0.49|
> |1-hop Test | 0.81 | 0.65 | 0.72 | 0.74 | 0.69 | 0.72 |
> |**1-hop Gap** | **0.39** | **0.25**| **0.23** | **0.38**|**0.31**| **0.23**|
> |2-hop Train | 0.52 | 0.55 | 0.54 | 0.37 | 0.40 | 0.36|
> |2-hop Test | 0.65 | 0.66 | 0.61 | 0.69 | 0.68 | 0.56|
> |**2-hop Gap** | **0.13**|**0.11** |**0.07** |**0.32** |**0.28** |**0.20** |
>
> **Q3:** Will ignoring odd-hop affect performance?
>
> **A3:** Even though EvenNet ignores odd-hop monomials in the spectral domain, this is not the case in the spatial domain.
> In fact, the aggregation of node $u$ preserves every one-hop neighbor that participates in a triangle with $u$.
> As shown in [1] and [2], such neighbors are common and important in social networks.
> For a triangle graph structure 'A1-A2-A3-A1' (they are of the same label), they are both the one-hop neighbor and the two-hop neighbor to others.
> Their information can be utilized by EvenNet, which is why EvenNet is still competitive on clean datasets.
>
> In Table 3 of the paper, even on clean and homophilic datasets Cora and PubMed, EvenNet outperforms vanilla GCN which heavily relies on first-order neighbors.
> Empirical evidence also suggests that EvenNet achieves results that are comparable to GPRGNN on clean datasets.

---

> ### Author Response · Authors · 2022-08-02
> **Our Response (II)**
>
> **Q4:** Theoretical analysis might not support the claim.
>
> **A4:** For Theorem 1, the average is taking over different homophily levels.
> MLP is usually not a good baseline compared with a well-learned graph filter.
> Yet, if the learned filter is going to be transferred to graphs of different homophily, we claim that MLP is a relatively good choice.
> For example, if a well-learned filter is applied to a completely random graph, there is no guarantee that the filter would outperform MLP.
> If the filter is applied to a graph of completely different properties, the overfitting of the training graph will introduce harmful information, which could make the filter worse than MLP.
>
> Take the results of MLP under DICE Attack in Figure 3 of the paper as an example.
> On the PubMed dataset, MLP generally outperforms advanced GNNs thanks to its lower variance.
> So we claim that on average, such an 'Even' operation won't hurt the average performance and is of lower variance when applied to graphs of different homophily, which is what we want to design as a robust spectral method.
> Of course, the perturbations of graph structure are usually not that severe that the modified graph is completely random, so our EvenNet still outperforms MLP in most cases.
>
> For Theorem 3, we are sorry for not making the assumptions clear.
> Theorem 3 tells that, if {$\alpha_i$} concentrate on different frequencies on training and test graphs, $L_o \sim$ $\sum (\alpha_i^2 - \alpha_{N-i}^2)g(\lambda_i)$ is small than $L_e$ on the test graphs, which means odd-order components contribute larger SRL in generalization.
>
> The key idea is to separate the odd and even parts of graph filters and see how they contribute to $L(G)$.
> The main difference between $g_{odd}$ and $g_{even}$ is that $g_{odd}$ is monotonic in [0, 2], while $g_{even}$ is symmetric about $\lambda=1$ and non-negative.
> This fact can be generalized to all polynomial filters.
> If we train an odd-order filter on homophilic graphs where $\alpha_i$ is generally larger than $\alpha_{N-i}$ (which means more frequencies components on low frequencies), $L_o$ is positive on the training graph with non-decreasing $g_{\lambda_i}$ and the SRL is minimized.
> Yet when transferred to a heterophilic graph, where $\alpha_i$ focus more on high frequencies, $L_o$ becomes negative and increases the SRL on the test graph.
> We will specify the assumption of Theorem 3 and make the proof clear in the revised version of our paper.
>
> **Q5:** Experiments on larger-scale datasets and real-world scenarios.
>
> **A5:** We didn't run experiments on large-scale datasets as the Meta / MinMax attacks are not scalable to large graphs.
> In **Appendix E \& F**, we have added defense results of EvenNet under Random attacks and scalable graph injection attacks on ogb-arxiv.
> Here we present the results and average training time per epoch of GCN, EvenNet, and EGCNGuard (An efficient GNNGuard from [3]) under Random attack on arxiv.
>
> | Methods \ Perturb Ratio} | 20\% | 40\% | 60\%|
> | :----- | :----: | :----: |:----: |
> |   GCN | 64.07  |  60.96  | 58.45 |
> |EGCNGuard | **64.52** | 60.81  | 57.18 |
> |EvenNet | 64.18 | **61.20**  | **58.97** |
>
> | Methods | Avg. training time per epoch (s$\times 10^{-3}$) |
> | :----- | :----: |
> | GCN | 0.253 |
> | EGCNGuard | 1.181 |
> | EvenNet | 0.38 |
>
> We can see that EvenNet is scalable to large graphs with low time consumption.
> We have added more detailed comparisons and experiments in **Appendix E** that demonstrate the efficiency of EvenNet.
>
> We report the homophily shift results under structure attacks on real-world datasets.
> We will look for more real-world scenarios besides structure perturbations in future work.
>
> If you still have any other questions about this work, we are looking forward to your reply!
>
> [1] Opsahl T, Panzarasa P . Clustering in Weighted Networks[J]. Social Networks, 2009, 31(2):155-163.
>
> [2] Ron Milo et al. “Network Motifs: Simple Building Blocks of Complex Networks” Science 2002
>
> [3] Yongqiang Chen, Han Yang, Yonggang Zhang, Kaili Ma, Tongliang Liu, Bo Han, James Cheng. Understanding and Improving Graph Injection Attack by Promoting Unnoticeability. ICLR 2022.

---

> > ### Comment · Reviewer_cSNX · 2022-08-05
> > **Response to Authors**
> >
> > Thanks for the authors' detailed responses and added experiments in such a short rebuttal time.
> >
> > For my first question “whether the methods could be effective to other types of attack not related to homophily change”, adding random perturbation is indeed a good candidate. But I might expect to see some more sophisticated methods.
> >
> > Take an example, in your current experiments, you use “include two poison attacks, Metattack (Meta) [37] and MinMax attack [30] with GCN the surrogate model”. With GCN as a surrogate model that is fragile to homophily changes, the attack model should also learn to change odd-order nodes. What if you use your proposed evennet as a surrogate model? Or considering some other attack model that specifically operated on evennet? (if you think taking your model as a target model is not very fair, you could consider a setting that for each GNN model, the attack model will take it as the target/surrogate).
> >
> > For the second question about considering heterophily graphs, I think there exist a lot of recent benchmarks for such association, including [1,2]
> >
> > [1] New Benchmarks for Learning on Non-Homophilous Graphs
> > [2] Graph Neural Networks for Graphs with Heterophily: A Survey
> >
> > A naive task could be a bipartite graph maybe? (For example, user-item network, where association happens at even-hop)
> >
> > I would like to see how well the proposed evennet performs on these datasets.
> >
> >
> > For the third question, it’s good to see evennet gets similar results as existing methods. Could authors add the number of parameters and also computational time for each method?
> >
> >
> > Overall, my major concern is still whether the proposed method could only handle a special type of graph. I could consider increasing my score if the authors provide more evidence that their methods could generalize to different types of graphs and attackers (especially for Non-Homophilous Graphs as well as non-homophilous attacks)

---

> > > ### Author Response · Authors · 2022-08-07
> > > **Our Response (III)**
> > >
> > > Thanks again for your insightful advice!
> > >
> > > **Q1:** Could EvenNet be the surrogate model? Why do we use GCN as the surrogate model?
> > > What about changing the setting for a more fair comparison?  For example, the attack model takes defense models as the target/surrogates.
> > >
> > > **A1:** We have added experiments about MinMax attacks on Cora/Citeseer/ACM datasets with GPRGNN and EvenNet being the surrogate models.
> > > We are unable to run Metattack with GPRGNN and EvenNet as the surrogate models due to its complexity in the meta-gradient calculation.
> > > Other experimental settings remain the same as those in the original submission.
> > > The results are presented in the below tables.
> > >
> > > **Defense results on Cora. Each row is of the same surrogate model.**
> > >
> > > | Surrogate \ Test Model | GCN | GPRGNN | EvenNet |
> > > | :-----  | :----: | :----: |:----: |
> > > |GCN    | 71.84 $\pm$ 0.91 | 77.32 $\pm$ 1.05 | **78.32 $\pm$ 0.81** |
> > > |GPRGNN | 81.09 $\pm$ 0.65 | 82.39 $\pm$ 0.75 | **82.70 $\pm$ 0.76** |
> > > |EvenNet | 81.80 $\pm$ 0.61 | 82.96 $\pm$ 0.69 | **83.08 $\pm$ 0.72** |
> > >
> > > **Defense results on Citeseer. Each row is of the same surrogate model.**
> > > | Surrogate \ Test Model | GCN | GPRGNN | EvenNet |
> > > | :-----  | :----: | :----: |:----: |
> > > |GCN | 68.57 $\pm$ 0.64 | 72.03 $\pm$ 0.70 | **73.38 $\pm$ 0.61**|
> > > |GPRGNN | 70.42 $\pm$ 0.48 | 72.23 $\pm$ 0.58 | **73.15 $\pm$ 0.66** |
> > > |EvenNet | 70.56 $\pm$ 0.57 | 72.03 $\pm$ 0.94 | **73.25 $\pm$ 0.90**|
> > >
> > > **Defense results on ACM. Each row is of the same surrogate model.**
> > > | Surrogate \ Test Model | GCN | GPRGNN | EvenNet |
> > > | :-----  | :----: | :----: |:----: |
> > > |GCN | 65.92 $\pm$ 3.73 | 84.74 $\pm$ 2.08 | **87.28 $\pm$ 0.69** |
> > > |GPRGNN | 79.38 $\pm$ 4.40 | 86.45 $\pm$ 0.88 | **86.91 $\pm$ 1.37** |
> > > |EvenNet | 74.67 $\pm$ 4.28 | 87.08 $\pm$ 0.81 | **87.62 $\pm$ 1.51** |
> > >
> > > EvenNet is the most robust model against various surrogate attacks, as shown by the above results. We can also observe that GPRGNN and EvenNet achieve even better defense performance when they are used as surrogate models rather than GCN.
> > > At first glance, the results appear counterintuitive; however, it should be noted that existing attack models require complex computation of the gradient of the adjacency matrix. Therefore, directly attacking spectral models presents an optimization challenge for the attack model.
> > > On the other hand, it is demonstrated that the simple perturbed graph with GCN as the surrogate applies to spectral GNNs.
> > > As a result, we follow most previous works from both attack and defense sides and choose GCN as the surrogate model in the original submission [1][2][3][4].
> > > We believe that developing attack models that target spectral GNNs is an exciting future direction.
> > >
> > > We did not consider the change of setting in the original submission because,
> > > on side of the attacker, it is usually not allowed to know the defense model ahead of the defense.
> > > And as we point out, it is hard for current methods to attack advanced GNN, such as GCNGuard and ProGNN, while still being efficient and effective.
> > > On the defender side, this new setting can be regarded as a Max-Min-Max optimization problem if we consider the attack process as a Min-Max optimization[2], which has been little discussed by previous works in terms of its complexity and also beyond the scope of our paper.
> > >
> > > **Q2:** Where the robustness of EvenNet comes from?
> > >
> > > **A2**:
> > > From a spatial standpoint, it is understandable that the scenarios that EvenNetcan be applied to seem constrained because it merely "ignores" the odd-order polynomial terms.
> > > However, information from odd-order neighbors is not entirely disregarded. (as we responded in **A3** in the preceding comment)
> > > More fundamentally, as a spectral method, EvenNet's robustness is explained by the stability of the obtained even-order filter, which is discussed in Theorem 1 when transferred to graphs of varying homophily and in **Appendix D** when confronted with more general perturbations.
> > > On the other hand, this paper concentrates on non-target graph structural attacks.
> > > While perturbations to the even-order neighbors of a single node may affect EvenNet's performance, it is difficult to perturb the even-order neighbors globally. (as stated in our response in **A2** in the previous comment)
> > > EvenNet continues to be superior in general defense tasks, especially during homophily generalization.
> > > For a better understanding, we will add specifications of the scope of EvenNet and intuitive examples.

---

> > > ### Author Response · Authors · 2022-08-07
> > > **Our Response (IV)**
> > >
> > > **Q3:** Could EvenNet still be effective when defense on non-homophilic graphs?
> > >
> > > **A3:** We have conducted additional MinMax attack experiments on two relatively large heterophilic datasets: Chameleon and Squirrel, with GCN being the surrogate model. The perturb ratio is set to be 20%.
> > > The results are summarized as follows:
> > >
> > > **MinMax attack on heterophilic graphs**
> > > | Method \ Dataset | Chameleon | Squirrel |
> > > | :-----  | :----: | :----: |
> > > |MLP |  48.84 | 30.31  |
> > > |GCN |  49.93 | 31.16 |
> > > |EGCNGuard |  45.33 | 27.34  |
> > > |H2GCN |  51.42 | 28.41 |
> > > |FAGCN | 49.98  | **33.64**  |
> > > |GPRGNN | 50.42 | 32.46  |
> > > |EvenNet | **52.87** | 33.21 |
> > >
> > > EvenNet remains to the most effective defense against attacks on heterophilic graphs, as demonstrated by the results. This credit is attributed to the fact that 1) Unlike GCN, which uses a fixed low pass filter, EvenNet can learn a more flexible graph filter, and 2) Unlike GPR-GNN, EvenNet can learn a more stable polynomial filter.
> > >
> > > As a supplement, we will include the MinMax results on the two heterophilic datasets.
> > >
> > > **Q4:** More details about the number of params of EvenNet and other baselines as well as the running time.
> > >
> > > **A4:** In **Appendix E**, we have included a discussion of the theoretical complexity of EvenNet and a comparison of the running time.
> > > Here, we add comparisons about the number of parameters and running time against more baselines.
> > > We report the number of parameters on dataset Cora and the average training time per epoch on dataset arxiv.
> > > We set the number of layers of each method to be **2**, the order of GPRGNN and EvenNet to be 10, and the size of
> > > hidden channels to be 64.
> > > The number of parameters and the average training time is summarized below:
> > >
> > > **Required Parameters on Cora**
> > > |Method | Parameters ($\times 10^3$) |
> > > | :-----  | :----: |
> > > |MLP |  92.2  |
> > > |GCN |  92.2  |
> > > |GPRGNN | 92.2  |
> > > |EvenNet | 92.2 |
> > > |GCNGuard | 92.4 |
> > > |FAGCN |  92.5 |
> > > |H2GCN |  95.2 |
> > > |EGCNGuard |  99.5 |
> > > |GCNII | 108.6 |
> > >
> > > **Average training time on arxiv**
> > > |Method | Avg. training time ($\times 10^{-3}s$) |
> > > | :-----  | :----: |
> > > |MLP | 0.168 |
> > > |GCN | 1.019 |
> > > |GCNII | 1.433 |
> > > |GPRGNN | 3.558 |
> > > |EvenNet | 3.688 |
> > > |FAGCN | 4.862 |
> > > |EGCNGuard | 9.384 |
> > >
> > > We see that generally, EvenNet takes up fewer parameters and is less time-consuming than other defensive models.
> > > Moreover, to achieve better performance, FAGCN / GCNII needs to use more layers in practice, which require more parameters and running time.
> > > In addition, methods such as H2GCN and GCNGuard are not scalable to large graphs for their space complexity.
> > >
> > > [1] Daniel Zügner, Stephan Günnemann: Adversarial Attacks on Graph Neural Networks via Meta Learning. ICLR, 2019
> > >
> > > [2] Kaidi Xu, Hongge Chen, Sijia Liu, Pin-Yu Chen, Tsui-Wei Weng, Mingyi Hong, Xue Lin:
> > > Topology Attack and Defense for Graph Neural Networks: An Optimization Perspective. IJCAI, 2019
> > >
> > > [3] Xiang Zhang, Marinka Zitnik: GNNGuard: Defending Graph Neural Networks against Adversarial Attacks. NeurIPS 2020
> > >
> > > [4] Wei Jin, Yao Ma, Xiaorui Liu, Xianfeng Tang, Suhang Wang, Jiliang Tang: Graph Structure Learning for Robust Graph Neural Networks. KDD 2020: 66-74
> > >
> > > [5] Yongqiang Chen, Han Yang, Yonggang Zhang, Kaili Ma, Tongliang Liu, Bo Han, James Cheng:
> > > Understanding and Improving Graph Injection Attack by Promoting Unnoticeability. ICLR, 2022
> > >
> > > [6] Qinkai Zheng, Xu Zou, Yuxiao Dong, Yukuo Cen, Da Yin, Jiarong Xu, Yang Yang, Jie Tang:
> > > Graph Robustness Benchmark: Benchmarking the Adversarial Robustness of Graph Machine Learning. NeurIPS Datasets and Benchmarks 2021

---

> > > > ### Comment · Reviewer_cSNX · 2022-08-07
> > > > **Response to Authors**
> > > >
> > > > Thanks so much for the author's prompt responses and added experiments.
> > > > My main concern of this paper is because it seems to have strong assumptions about the graph as well as the graph attacker, so I’m very concerned “what is the scope of application for the proposed method, and in what situation it might fail to work”. The experiments with EvenNet as a surrogate model and the experiments on non-homophily graphs solve part of this concern. I raise my score to 5 and lean towards accepting this paper.
> > > >
> > > > I still have several comments regarding the unsolved concerns:
> > > >
> > > > 1. The discussion about “EvenNet didn’t throw away all odd-order neighborhood information” in the spatial domain is very important to understand the contribution of this paper and why the proposed method works. I would recommend the authors add these discussions to the paper, and it would be better if you could conduct some quantitative analysis (for example, in real-world graphs, what is the correlation of the Evennet’s filter to the odd-order neighbors, e.g., first-hop neighbors, similar to the aggregation weights in GAT). Also, it might be good if the authors could have a piece of pseudocode in the paper to show how to really implement this method, in order to avoid misunderstanding. (especially given that the authors didn’t provide the code)
> > > >
> > > > 2. I understand the non-target attack setting is a reasonable setup, but still using the spatial GCN as a surrogate seems unfair. The added experiment that takes EvenNet as a surrogate model, as well as random perturbation, addresses this concern a lot, but as the authors pointed out, the MinMax attacker might not be a good choice for spectral GNN. Probably a more general method such as the one discussed in [1] would solve this problem?
> > > >
> > > > 3. If the authors think the proposed method is very general and could generally improve robustness in many different setups, the experiments need to consider a lot more different settings apart from only the non-target attack. Otherwise, it would be better to clearly state the assumption of this method, in what situation it could work, and is there any setting or some particular types of graph this proposed method could not handle, I think the authors could state them in the limitation part, and it won’t affect the contribution of this work as it already proved to be effective in non-target setting, but make the scope much more clear.
> > > >
> > > > [1] Adversarial Attack on Graph Structured Data

---

> > > > > ### Author Response · Authors · 2022-08-08
> > > > > **Our Response (V)**
> > > > >
> > > > > Thanks for supporting our work.
> > > > > In light of your suggestions, we'll add the following to the final version of the paper.
> > > > >
> > > > > (1) To clarify that EvenNet does not completely disregard information from odd-hop neighbors, we will add the example in "Our Response (I)" regarding the spatial domain performance of EvenNet.
> > > > > To better illustrate the robustness of even-order neighbors in the spatial domain, we will also include the homophily-gap table in "Our Response (I)".
> > > > > The advice regarding the quantitative analysis of odd-order neighbors is hugely helpful. In the final version of the paper, we will try to include some quantitative analysis.
> > > > > We have created an anonymous repository in the following dropbox link:
> > > > > https://www.dropbox.com/scl/fo/wbcp83v52h8au8fbx5dve/h?dl=0&rlkey=peo57et1hsehn1vdn9sz6u2ds, where we will upload our code by August 12, 2022.
> > > > >
> > > > > (2) To our knowledge, the mentioned RL-S2V is a targeted model based on reinforcement learning, which is not our focus and may still face optimization challenges when attacking spectral GNNs.
> > > > > That said,  we will attempt to include this attack model in the final version of the paper.
> > > > > Thanks for providing more attack methods.
> > > > >
> > > > > (3) We will address the scope of our method that we focus on non-target structural attacks in the final version.

---

> > > > > > ### Comment · Reviewer_cSNX · 2022-08-09
> > > > > > **Response**
> > > > > >
> > > > > > The RL-S2V, as well as the genetic algorithm baseline, only takes the output of GNNs without accessing their parameters, so I don't think there are "optimization challenges". It's okay that the authors focus on non-targeted attacks and don't consider RL-S2V, but you may want to make your scope clear.

---

> > > > > > > ### Author Response · Authors · 2022-08-10
> > > > > > > **Our Response (VI)**
> > > > > > >
> > > > > > > Thanks for your comment about the genetic algorithms!
> > > > > > > We will limit the scope of our method to non-targeted attacks in the final version.
> > > > > > > We are also trying to run RL-S2V, which could take some time in training.
> > > > > > > If we are able to report the results on time, we will add them to the dropbox link.

---

### Author Response · Authors · 2022-08-08
**Our Response to all Reviewers**

We sincerely thank all of the reviewers for your time and efforts.

We have built an anonymous repository in the following dropbox link:
https://www.dropbox.com/scl/fo/wbcp83v52h8au8fbx5dve/h?dl=0&rlkey=peo57et1hsehn1vdn9sz6u2ds, where we will upload our code by August 12, 2022.

---

### Meta-Review · Area_Chair_nyQa · 2022-08-24

**Recommendation:** Accept
**Confidence:** Less certain

**Metareview:**

This paper proposes a simple and effective idea to only use even-order neighbors to improve the robustness and generalization ability of spectral GNNs. It is based on the intuition that a friend's friend is friend, and an enemy's enemy is also friend, thus only using even-order neighbors improves the generalization across different homophily/heterophily levels. Considering that spectral GNNs' expressive power have been shown to easily saturate recently, analyzing their generalization power is of great importance and is a natural next step. All the reviewers agree on the value of the paper. However, they also point out several issues, such as concerns on the theoretical analysis. I encourage the authors to further polish the paper and improve the theoretical analysis in the camera ready version.

**Award:**

No

---

### Decision · Program_Chairs · 2022-09-14

Accept